



# Early Holocene cold snaps and their expression in the moraine record of the Eastern European Alps

Sandra M. Braumann[1,2], Joerg M. Schaefer[2], Stephanie M. Neuhuber[1], Christopher Lüthgens[1], Alan J. Hidy[3], Markus Fiebig[1]

[1]University of Natural Resources and Life Sciences (BOKU), Peter Jordan-Str. 82, 1190 Vienna, Austria
[2]Lamont-Doherty Earth Observatory of Columbia University, Division of Geochemistry, Palisades, NY 10964, USA
[3]Center for Accelerator Mass Spectrometry, Lawrence Livermore National Laboratory, Livermore, CA 94550, USA

*Correspondence to*: Sandra M. Braumann ([sandra.braumann@boku.ac.at](mailto:sandra.braumann@boku.ac.at))

**Abstract.** Glaciers preserve climate variations in their geological and geomorphological records, which makes them prime candidates for climate reconstructions. Investigating the glacier-climate system over the past millennia is particularly relevant because, first, the amplitude and frequency of natural climate variability during the Holocene provides the climatic context against which modern, human-induced climate change must be assessed. Second, the transition from the last glacial to the current interglacial promises important insights into the climate system during warming, which is of particular interest with

respect to ongoing climate change.

Evidence of stable ice margin positions that record cooling during the past 12 ka are preserved in two glaciated valleys of the Silvretta Massif in the Eastern European Alps, the Jamtal (JAM) and the Laraintal (LAR). We mapped and dated moraines in these catchments including historical ridges using Beryllium-10 Surface Exposure Dating ([10]Be SED) techniques, and correlate resulting moraine formation intervals with climate proxy records to evaluate the spatial and temporal scale of these cold phases.

The new geochronologies indicate two moraine formation intervals (MFI) during the Early Holocene (EH): 10.8 ±0.7 ka (n=9) and 11.2 ±0.8 ka (n=12). Boulder ages along historical moraines (n=6) imply at least two glacier advances during the Little Ice Age (LIA; c. 1250-1850 CE), around 1300 CE and in the second half of the 18[th] century. An earlier advance to the same position may have occurred around 500 CE.

The Jamtal and Laraintal moraine chronologies provide evidence that millennial scale EH warming was superimposed by

centennial scale cooling. The timing of EH moraine formation is contemporaneous with brief temperature drops identified in local and regional paleoproxy records, most prominently with the Preboreal Oscillation (PBO), and is consistent with moraine deposition in other catchments in the European Alps, and in the Arctic region. This consistency points to cooling beyond the local scale and therefore a regional or even hemispheric climate driver. Freshwater input sourced from the Laurentide Ice Sheet (LIS), which changed circulation patterns in the North Atlantic, is a plausible explanation for EH cooling and moraine

formation in the Nordic region and in Europe.



## 1 Introduction

The transition from the Younger Dryas (YD; 12.9–11.7 ka) to the Holocene (c. 11.7 ka to present) is an important period for studying the climate system, its forcings and its feedbacks. Climatic conditions shifted from glacial to full interglacial conditions within approximately two millennia, between 12 and 10 ka (e.g. Alley, 2000;Marcott et al., 2013;Rasmussen et al.,

2006). This general warming trend was interrupted by abrupt centennial scale cooling that appears to be linked to freshwater input into the North Atlantic (e.g. Thornalley et al., 2010;Bjorck et al., 1997;Hald and Hagen, 1998;Nesje et al., 2004). The climatic shift from the YD to the Early Holocene (EH) was accompanied by a multitude of major environmental changes that are interconnected, including the melting of ice caps and glaciers in both hemispheres, changes in the atmospheric composition and in circulation patterns, and the reorganization of ocean currents (e.g. Clark et al., 2012;Shakun et al., 2015;Denton et al.,

2021). Human-induced warming since the beginning of the industrial era is on a trajectory to lead to changes of similar magnitude in our environmental system, yet at an even faster pace (Gobiet et al., 2014;Beniston et al., 2018). By investigating the timing of YD-EH warming and its perturbations, we can broaden our knowledge on natural drivers and physical mechanisms, which modulated the climate system at that time. Information on climate oscillations gained from this major natural transition from glacial to interglacial conditions build a valuable foundation for disentangling natural and anthropogenic

forcings, and their respective relevance, especially in the light of the ongoing transition from an interglacial to an industrialized world.

Glaciers respond to climate fluctuations sensitively and are important elements for understanding the climate of the past (Roe et al., 2017;Huston et al., 2021). Reconstructing former ice margins allows deciphering glacier fluctuations across time and

space and informs us on climate variations that drove these changes. Mountain glaciers in alpine, melt-dominated regimes are most sensitive to changes in summer temperature, to a lesser extent to changes in precipitation (e.g. Oerlemans, 2005;Steiner et al., 2008;Rupper and Roe, 2008). At the end of the Late Glacial (LG), YD cooling resulted in glacier stabilization or readvance in the European Alps and lead to the deposition of moraine sets, whose estimated Equilibrium Line Altitudes (ELA) are approximately 250 to 350 m below ELAs of glaciers during the LIA (e.g. Ivy-Ochs, 2015). These moraines are termed

'Egesen' moraines and have been subject of numerous cosmogenic nuclide studies, which have advanced our understanding of glacier responses to cooling during the Late Glacial (LG) (e.g. Kelly et al., 2004;Ivy-Ochs et al., 2009;Federici et al., 2008;Kerschner and Ivy-Ochs, 2008;Cossart et al., 2012;Ivy-Ochs et al., 2006). In parallel, first attempts had been made to produce direct ages of younger moraines that were identified inboard the presumable Egesen moraines, but outboard historical LIA margins (Kerschner et al., 2006;Ivy-Ochs et al., 2006). Based on their morphostratigraphy, these moraine ridges were

postulated as type localities for preboreal glacier advance, for instance the Kartell moraines in the Verwall area and the Kromer moraines in the Silvretta Massif, both in the Eastern Alps (e.g. Gross et al., 1978;Faedrich, 1979). Kartell moraines are today placed into the latest YD (Egesen-III); recalculated [10]Be ages of Kromer moraines suggest moraine deposition during the EH, around 10 ka (Ivy-Ochs et al., 2006;Kerschner et al., 2006;Moran et al., 2016b). Dating efforts that address presumable EH





moraines continued toward the Central and Western Alps and have produced chronologies, which substantiate moraine
formation between 12 and 10 ka, although not necessarily synchronously (Protin et al., 2019;Protin et al.,
2021;Schimmelpfennig et al., 2012;Schimmelpfennig et al., 2014;Boxleitner et al., 2019b;Schindelwig et al., 2012;Moran et
al., 2016b;Moran et al., 2016a;Moran et al., 2017a;Hofmann et al., 2019;Baroni et al., 2017;Cossart et al., 2012;Boxleitner et
al., 2019a). In a few recent studies this pattern of moraine deposition has been confirmed in the Eastern Alpine region (Moran
et al., 2016a;Bichler et al., 2016;Moran et al., 2017b;Moran et al., 2017a). The youngest dated EH moraine is located in the
Ochsental, a valley adjacent to the sites discussed in this study (Braumann et al., 2020). The relevance of this chronology lies
in the finding that glaciers in the valley had melted back to historical sizes around 10 ka, and that they have remained within
these limits throughout the Holocene, which is consistent with complementary paleoproxy records from the Eastern Alps (e.g.
Patzelt, 2016;Dietre et al., 2014;Nicolussi and Patzelt, 2000).

To intensify our knowledge on EH glacier configurations in the Eastern Alps, where directly dated moraine ages remain sparser
compared to the Western and Central Alps, we conducted a geochronological study in two glaciated catchments in the Silvretta
region in the Eastern Alps. We applied state of the art cosmogenic nuclide techniques to date moraines inboard presumable
LG ice margins to constrain the timing of Holocene cold phases recorded in the moraine record. We chose this region for two
reasons: First, moraine sets which postdate the LG phase are well-preserved in the Silvretta Massif and show a multi-ridge
structure, which promises insights into repeated Holocene cooling at times, when glaciers were larger than during the LIA.
Second, in addition to comparable cosmogenic nuclide records in the region (Moran et al., 2016b;Braumann et al., 2020;Ivy-
Ochs et al., 2006), high-quality paleoenvironmental, archeological and historical information on Holocene climate, which
complements the moraine record, is available (Kasper, 2013;Dietre et al., 2014;Nicolussi, 2010;Patzelt, 2019;Kasper, 2015).
The primary objective of this study is to generate more detailed and robust moraine chronologies in the Eastern Alpine region,
which contribute to our understanding of the spatial and temporal pattern of glacier advances throughout the Holocene with
an emphasis on the EH. We correlate the new moraine chronologies with moraine records and climate proxy data from the
Alpine region and from other glaciated regions in the Northern hemisphere, and identify climate signals that are coherent with
Holocene glacier and climate evolution in the Silvretta Massif. Finally, we discuss possible links between climatic trigger
events and EH cold snaps, which manifest in the moraine record of the Northern hemisphere.

**2 Setting**

The study sites are located at the north-facing side of the Silvretta Massif, a mountain range in the transition zone between the
Eastern and Western European Alps, at the border of Austria and Switzerland (Fig. 1a). Moraine sets from two adjacent valleys,
the Jamtal (JAM) and the Laraintal (LAR), were investigated and used for glacier reconstructions. Both valleys are north-south
oriented, drain northwards into the Danube catchment and are at present glaciated only in their highest sections (>2400 m
a.s.l.) (Fig. 1b). The main and most prominent glacier of the Jamtal is the Jamtalferner with an area of c. 2.8 km² (DEM of





2018). Smaller glaciers such as the Totenfeld, the Getschnerferner and the Augustenferner have retreated to cirque positions and are not connected to the valley glacier today (Fig. 1c). The situation is different in the neighboring Laraintal, where the Larainferner, which covers c. 1 km² (DEM 2018), is the only glacier still present in the valley (Fig. 1d).


**Figure 1: Geographic location and lithology of investigated glaciated catchments. (a) Central Europe with the Alpine mountain range highlighted in dark grey and Austria outlined with black line. The Silvretta Massif (star symbol) is located in the westernmost part of the Eastern Alps. (b) North-facing section of the Silvretta Massif. Blue shading illustrates glacier outlines in the year of 2012 (Fischer et al., 2015). Jamtal (JAM) and Laraintal (LAR) are subject of this study. Moraine chronologies of Kromertal (KR),**
**Klostertal (KL) and Ochsental (OcG-GrK) will be discussed later in this article (Moran et al., 2016b;Braumann et al., 2020). Red circles mark locations of subfossil tree findings. 1 – Bielerhöhe, 2 –Klostertal (Nicolussi, 2010), 3 – Futschöltal (F) (Patzelt, 2019), 4 – Las Gondas (Nicolussi, 2010;Dietre et al., 2014) (c). Updated geological and geomorphological maps of Jamtal and (d) Laraintal, modified from Fuchs and Oberhauser (1990). Viewpoints from which photos in Figs. 3 and 5 were taken are denoted with white-blue symbols. Digital Elevation Model (DEM) provided by © Land Tirol (resolution 1 m).**

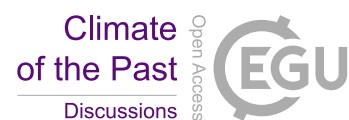

The closest meteorological station recorded a mean annual atmospheric temperature of 3.1° C averaged over the period 1981–
       2010 (station number 101949; 1587 m a.s.l.) (BMNT, 2016). The mean annual precipitation measured at the same station
       amounts to 1087 mm/yr for the same reference period, with a snow cover present on 175 days/yr on average. Precipitation is
       also measured in proximity to the Jamtalferner tongue (2400 m a.s.l.) and yields an annual mean of 1507 mm/yr (reference
       period 1989–2017) (Fischer et al., 2019). Since the end of the LIA, all Silvretta glaciers have retreated in response to almost

continuous warming and have lost about two thirds of their areas relative to the LIA maximum (Fischer et al., 2021;WGMS,
       2018). Geodetic mass balance estimates of Silvretta glaciers document increased losses within recent decades (Fischer et al.,
       2021). While geodetic mass balance averaged across all Silvretta glaciers amounted to -0.2 ±0.1 m w.e./yr in the reference
       period from 1969–2002, this value increased to -0.8 ±0.2 m w.e./yr between 2006–2018. For Jamtalferner and Larainferner,
       mass losses from 2006–2018 are quantified to -1.0 ±0.2 m w.e./yr and to -0.8 ±0.2 m w.e./yr, respectively.

The Silvretta Massif contains some of the oldest rocks of the Eastern Alps with a presumed depositional age in the Precambrian
       followed by several metamorphic events (Maggetti and Flisch, 1993;Bertle, 1973). Lithology in the region consists of
       crystalline rocks, which are part of the Upper Eastern Alpine tectonic unit, more precisely the Silvretta-Seckau nappe (Schuster,
       2015;Fuchs and Oberhauser, 1990). Rocks at the Jamtal and Laraintal formed during the Permian, experienced repeated
       faulting prior to and throughout the Alpine orogeny and are thus highly metamorphic (Friebe, 2007 and references therein).

Predominant rock types in the study area are different gneiss variations and amphibolites (Fig. 1c and 1d). Rock samples that
       were collected from moraines yield quartz contents – the target mineral for the applied cosmogenic nuclide method – between
       0.3 % and 26.1 % with a median of 3.9 % (Table S1).

## 3 Methods

### 3.1 Principle of $^{10}$Be surface exposure dating

Glaciers erode into bedrock and transport rock material to their margins. When glaciers are stationary for several years or
       longer, linear landforms – moraines – that consist of glacial debris, accumulate at their ice margins. Dating these moraines
       unravels the timing of glacier stabilization or rather the beginning of glacier retreat, and allows the reconstruction of glacier
       configurations of the past. For $^{10}$Be SED – the cosmogenic nuclide approach used in this study – rock samples were extracted
       from boulder surfaces deposited along moraines. These boulders were carved out of bedrock by glacial flow and were for the

first time exposed to high-energetic cosmic radiation when they melted out of glacial ice. Secondary cosmic rays interact with
       Si and O in quartz and produce radionuclides, among others $^{10}$Be (Lal, 1988). The annual production rate of $^{10}$Be is well
       constrained today and the accumulation of the radionuclide is used to determine the duration of exposure by quantifying $^{10}$Be
       in rock surfaces of moraine boulders (Gosse and Phillips, 2001).



## 3.2 Geomorphological mapping and rock sample collection

We build upon geological and geomorphological maps, which were produced in previous studies and which were the basis for further detailed field investigations in the years of 2018 to 2020 (Fuchs and Oberhauser, 1990;Hertl, 2001;Fischer et al., 2015;Fischer et al., 2019). In the course of a general survey of the Jamtal and the Laraintal area, we complemented and updated preexisting maps according to our own mapping. We then focused on the mapping of glacial features and placed particular emphasis on the fine structure of historical moraines and ridges that were identified outboard historical moraines. The dating

of these structures promises to shed light onto the timing of climate perturbations, which favored moraine formation when glaciers were still relatively large compared to their present-day configurations. In order to ensure robust landform age calculations, we took three or more rock samples from each selected ridge, provided that they fulfilled our sample selection criteria described in detail in Braumann et al. (2020; Appendix S-Table 1). In total, 27 samples were extracted from boulders using hammer and chisel, and an electric saw. Geographic coordinates of sampled boulders were measured using a hand-held

GPS device. Strike and dip of sampled surfaces were quantified with a geological compass. Sample elevations were taken from the DEM of 2018 (x-y-z resolution 1 m, © Land Tirol). Shielding was calculated using the ArcGIS 'Skyline' Toolbox.

## 3.2 Sample preparation and age calculation

All samples were processed at the Cosmogenic Isotope Laboratory of the Lamont-Doherty Earth Observatory (LDEO) following the geochemical standard protocol for quartz preparation and the extraction of $^{10}$Be (Schaefer et al., 2009;LDEO,

2012b, a). Prior to quartz digestion using concentrated hydrofluoric acid, approximately 180 µg of the LDEO $^{9}$Be carrier made of deep-mine Beryl was added to the samples (carrier concentration c. 1000 ppm). Samples LAR-19-14 and LAR-19-16 with extremely low quartz contents of 0.3–0.4 %, equivalent to a yield of purified quartz of c. 2.6–2.7 g per sample, were treated differently. Due to their small sample sizes combined with our EH age estimates, we expected low total numbers of cosmogenic $^{10}$Be atoms in the samples. For these samples we adopted a sample preparation procedure where $^{9}$Be carrier is reduced and

replaced with Fe carrier. Only c. 100 µg $^{9}$Be carrier was added during digestion and then c. 100 µg of Fe (concentration 1000 mg/L) was added to the samples prior to hydroxide precipitation (as Be(OH)$_2$ + Fe(OH)$_2$), subsequent to the cation columns step. The Fe addition allowed us to maintain a manageable sample volume, which facilitates the handling of the samples. This procedure was recommended for exceptionally low-level samples based on unpublished experimental data that suggests it optimizes $^{10}$Be counting efficiency at the Center for Accelerator Mass Spectrometry (CAMS) facility, Lawrence Livermore

National Laboratory (LLNL) (pers. comm. A. Hidy). We proceeded with subsequent steps of sample preparation according to the LDEO protocol (LDEO, 2012b). Isotope ratios ($^{10}$Be/$^{9}$Be) in samples were measured at CAMS-LLNL using the 07KNDSTD3110 standard with a $^{10}$Be/$^{9}$Be ratio of 2.85×10$^{-12}$ (Nishiizumi et al., 2007).

Exposure ages were calculated using the online calculator formerly known as the CRONUS-Earth online calculator (v3) (Balco

et al., 2008). We applied the regional Swiss production rate (Claude et al., 2014), and 'Lm' scaling to account for site specific



nuclide production. All [10]Be boulder ages are arithmetic mean ages presented with 1σ analytical uncertainties, including a 1 % uncertainty on the carrier concentration. Moraine ages represent arithmetic means of exposure ages of three or more boulder ages. Uncertainties reported with moraine ages include the production rate uncertainty (for the Swiss production rate c. 6.3 %) in addition to the analytical and carrier uncertainty, and were propagated in quadrature. Identification of potential outliers was

accomplished following the $\chi^2$ statistics implemented in the online calculator.

Corrections of exposure ages for seasonal snow cover were not applied. If exposure ages were significantly influenced by snow effects, age dispersion would be expected among boulders embedded in the same moraine, but whose shapes and exposures vary. Our data do not show a significant bias of this type; therefore, snow cover effects appear to be insignificant at our study sites (see results section, and Supplements, sect. 4 and 5).

The preservation of striations on rock surfaces and the general condition of boulder surfaces in the valleys suggest that erosion has not significantly impacted their surfaces since deposition. Therefore, all ages that are presented and discussed in the following represent values without any erosion correction applied. However, in some studies addressing the Holocene time scale, an erosion rate of 1 mm/ka is used (André, 2002). To test the sensitivity of our ages to this erosion rate, we recalculated our data using this value and find that ages become at most 1 % older (median 0.8 %; Table S3), an age shift that is not

significant on the 1σ level and that does not change our interpretation of the data.

## 4 Results

### 4.1 Geomorphology

In both valleys, distinct moraine sets, which mark Holocene paleo-ice margins, are preserved. We numbered the moraines from the youngest J0 to the oldest J5 at the Jamtal, and moraines at the Laraintal from L1 to L5 in analogy.

**4.1.1 Jamtal**

At the Jamtal, the innermost moraine we address in this article, is **J0**. The moraine was deposited at the left-lateral valley flank, inboard the presumable LIA moraine (**J1**; Fig. 2). J0's age of deposition falls into the period between the end of the LIA and the turn of the 20[th] century Fischer et al. (2019). J0 and J1 are punctuated by a c. 100 m wide drainage channel, which evolved along the flowline of a former tributary glacier (Totenfeld, Fig. 3a and A1c). With numerous bedrock outcrops along the valley

flank and a slope of >35° the terrain is steep and impedes the accumulation moraines higher up (Fig. 3b). On the right-lateral side, in turn, slope angles are lower (c. 5-35°) and allowed the formation and preservation of multi-ridge moraine complexes (J1 to J4; Fig. 2, A1d and A1f). J1 consists of fresh, blocky debris with little to no lichen colonialization. Pioneer plants grow in voids in between blocks, whereas segments with more fine sediment the ridge is covered with a thin vegetation layer (Fig. A1a–c). In some sections, J1 is several tens of meters wide, which contrasts with the adjacent J2 with a maximum width of

about 8 m. J2 is located in a depression between J1 and a till covered slope, on top of which J3 and J4 were deposited (Fig. A1e–f). J2 is rich in fine sediment but does not feature boulders that meet our [10]Be sample selection criteria.







**Figure 2: Holocene moraine chronology of Jamtal. J0 (red) has been deposited after the LIA but prior to the 20th century (Fischer et al., 2019). J1, J3R and J4R were dated in this study. Ages along the J1 moraine (pink) indicate that Jamtalferner reached its historical maximum during the second half of the 18th century, and during the Neoglacial (NG), c. 500 CE. Earlier phases of glacier stabilization that exceeded subsequent Holocene culminations, are evidenced by moraines J3R and J4R, which both date to the EH.**





**Figure 3: Photographs of Jamtal. (a) View toward Jamtalferner and Totenfeld. ¹⁰Be sample locations are marked with colored circles (pink – LIA; black – Neoglaical, cyan and blue – EH). (b) Downvalley view depicting sample locations JAM-18-06 (pink) and JAM-18-07 (black). (c) Valley flank below Chalausferner and Augustenferner with sampled boulders along the innermost (pink), the middle (cyan) and the outer (blue) right lateral moraine of former Chaulausferner. For the broader context of the individual sites, see Figure 1c.**


**J3** and **J4**, two parallel, curved moraines about 20 to 30 m further uphill relative to J2, are right-lateral moraines of Chalausferner, and evidence the convergence of this tributary glacier and the Jamtalferner (Fig. 3c). Boulder surfaces embedded in these moraines are populated with black and green lichens and show signs of weathering, for instance cracks and exfoliation. On the valley floor, a moraine with a frontal position at an altitude of c. 2120 m a.s.l. is preserved. The ridge

consists of weakly weathered material and is in this respect as well as with respect to geometry the terminal equivalent of the





lateral **J1** moraine (Fig. 2 and A2c). This correlation is in accordance with glacier outlines of the Austrian Glacier Inventory (AGI; Fischer et al., 2015), with a geomorphological map compiled by Hertl (2001:226) and with results from a recent study on vegetation dynamics at the Jamtal, which includes ice margin reconstructions since the end of the LIA (Fischer et al., 2019).

North of the terminal section of J1 is an area covered with angular and subangular blocks, whose surfaces are significantly more weathered compared to J1, and which exhibit extensive lichen population (Fig. A2c–e). Many blocks have cracks and are fractured, which may indicate impacts associated with gravitational movement. The morphology outboard J1 is convex – unusual for in situ rockfall deposits, which typically form lobate structures with large boulders in frontal positions. We therefore hypothesize that these deposits stem from rock failures along one (or both) valley flank(s) farther uphill. The material collapsed onto the formerly larger glacier and was transported downstream through glacial flow. As the glacier retreated, these

rockfall deposits melted out and accumulated on the valley floor. An additional argument supporting this scenario is the provenance area of a potential rockfall event, which could not clearly be identified along the surrounding walls and peaks. The blockfield was in part overprinted by one (or multiple) glacier advances, as evidenced by the position of moraine J1.

A set of ridges in the right latero-frontal section outboard the J1 moraine appears to be somewhat displaced (Fig. 2 and A2f). A scarp above this moraine set and a stabilized sliding mass below caused an offset of formerly connected crests. Together

with rockfall deposits from bedrock outcrops in higher up sections, these moraines are not considered as prime candidates for [10]Be sampling. We also avoided structures close to the Jamtal hut. Even though we identified several ridges in its vicinity, which were presumably deposited during the Holocene, land surfaces in this area have been anthropogenically altered. For instance, the road leading up to the hut and the hut itself are built on moraines (Hertl, 2001:77). The same argument applies to a ridge at an elevation of c. 2045 m a.sl. denoted as **J5** in (Fig. 2 and A2a–b). Although the structure may be interpreted as a

Holocene terminal moraine it was rejected for sample collection as it is in part anthropogenically overprinted and may comprise boulders disintegrated from the right-lateral slope above.

Evidence of older, LG terminal positions farther downstream is scarce. Hertl (2001:77) describes a lineament that dips to the valley floor about 2.5 km downstream of J5, at an elevation of 1900–1920 m a.s.l. The author tentatively interprets the structure as a latero-frontal moraine deposited towards the YD termination. C. 8 km downvalley from J5, a tripartite moraine set

(Gaffelar settlement) is attributed to an earlier YD phase, yet not the YD maximum. Lateral LG moraines are absent in the main valley, but are preserved in the Futschöl tributary valley, which joins the Jamtal from the East in the area of the Jamtal hut (Fig. 1b).

### 4.1.2 Laraintal

At the Laraintal, we focused on valley sections outlined in Figure 1d and detailed in Figure 4. Texture, relative positions, and

structure of moraines in this valley resemble moraine sets at the Jamtal. L1 – the presumable LIA ridge – consists of fresh, sparsely vegetated debris and is traceable along both valley flanks. On the eastern side, we identified a fine-structured set of moraines which we refer to as L2, L3R and L4R in Figure 4. Similar to J2, L2 with a width of c. 8–10 m is less prominent compared to L1 (>20 m; Fig. A3b). Also, L2 has a high fine sediment content with no large boulders on the crest. J3R has a



broader, but less pronounced crest, followed by J4, the outermost moraine in this valley section (Figure 5c). Boulders of both
moraines, J3R and J4R, were sampled for ¹⁰Be extraction. Vegetation cover has developed on the surfaces of both ridges and
soil formation processes are advanced on the glacier-distal side of J4R.

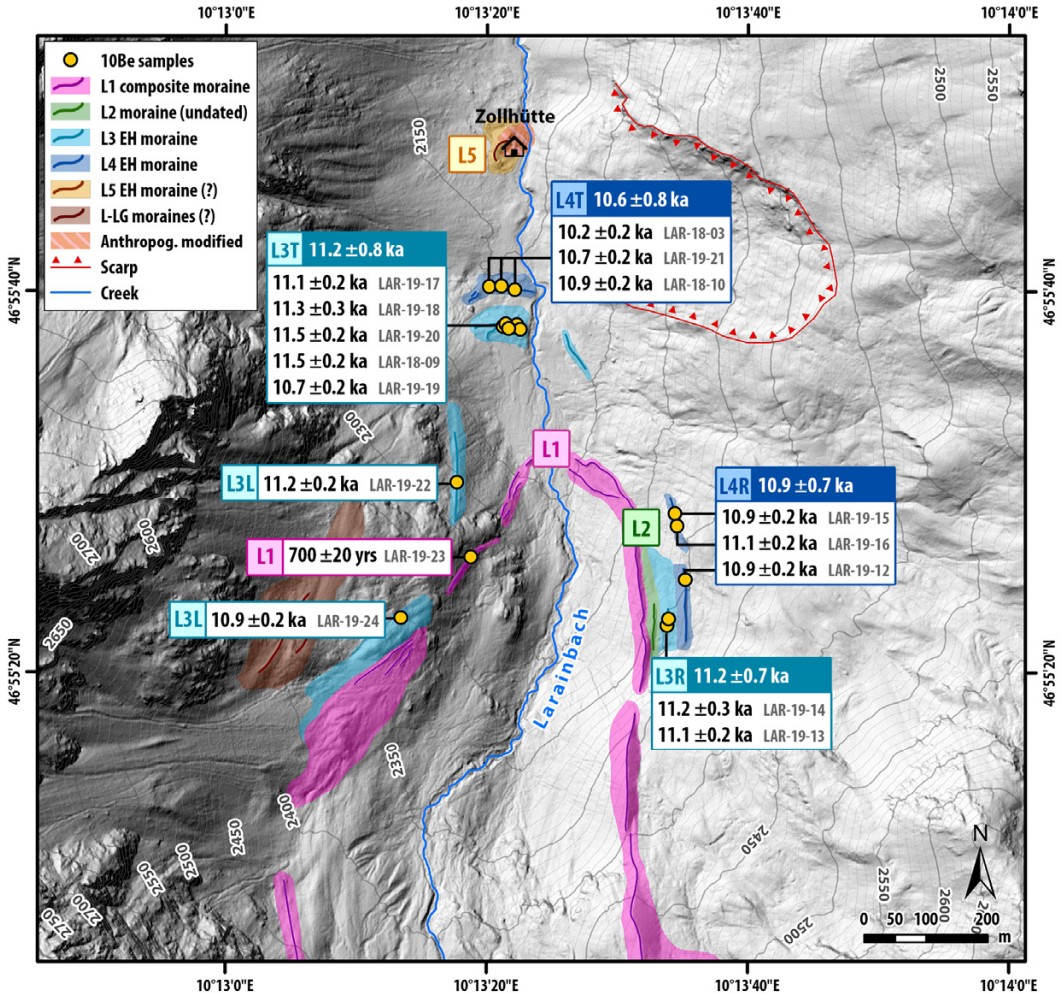

Figure 4: Holocene moraine chronology of Laraintal with moraine ages displayed. The ¹⁰Be sample collected from the L1 indicates
an early LIA advance around 1300 CE. Lateral and terminal moraines outboard L1 yield EH ages, which agree well with the Jamtal
moraine record.



**Figure 5: Photographs of Laraintal. (a) Terminal moraine section with sample locations marked with circles. (b) Left lateral valley flank with LAR-19-22 (cyan), LAR-19-23 (pink) and LAR-19-22 (cyan) from left to right. (c) Right lateral moraine set with LAR-19-15 in blue in the foreground, and LAR-19-13 and LAR-19-14 in the background (cyan). For the broader context of the individual sites, see Figure 1d.**

Approximately 200–250 m further downstream, at an elevation of 2130 m a.s.l., is another set of ridges, on one of which a small hut ("Zollhütte") was built (Fig. 4). This Zollhütte ridge (L5) is framed by a block field consisting of a blend of angular and rounded boulders. A massive debris cone west of Zollhütte, which has a layer of coarse and medium-sized, fresh material





on top, points to continuous sediment supply from the left-lateral wall. To the east, a scarp with a concave surface below indicates former (and possibly ongoing) sliding processes directed towards Zollhütte. Moreover, rockfall events with material disintegrating from the wall below the "Hoher Kogel" peak have been witnessed during field work in the year of 2019, with boulder volumes of multiple cubic meters that have crashed on the valley floor (Fig. 1d and A4a–b). Such events have probably

also occurred in the past as the wall exhibits multiple lighter sections, which are indicative of removed material and thus of previous rock failures. Due to the manifold processes, which impacted the Zollhütte area, the J5 ridge is risky to tackle with SED of boulders, even though it probably delimits a terminal glacier position.

### 4.2 [10]Be results

[10]Be analytical data of all 27 boulder samples and corresponding age information are listed in Table 1 (Jamtal) and Table 2

(Laraintal). Kernel plots of moraine ages are displayed in Figure 6 (LIA) and Figure 7 (EH). Ages are reported for each valley individually and are discussed according to their landform number in ascending order, from J1 to J4 and from L1 to L4. All exposure ages fall into three periods of high(er) glacier activity within the past 12 ka: the LIA, the first millennium common era, and the EH.

### 4.2.1 Jamtal

Five samples were collected from boulders along moraine **J1** (Fig. 2, Table 1). Three of them (JAM-18-06, JAM-18-17, JAM-18-18) were deposited during the second half of the 18th century and yield a rounded mean age of **260 ±20 yrs** (Fig. 6). The other two samples, JAM-18-07 (1500 ±50 yrs) and JAM-18-16 (1070 ±20 yrs), both produce neoglacial ages. Since J2 lacks suitable boulders for [10]Be sampling, the subsequent dated ridge is J3R. Based on three boulder ages of moraine **J3R** (JAM-18-01: 11,020 ±200 yrs, JAM-18-02: 11,280 ±180 yrs, JAM-18-03: 11,850 ±220 yrs), we calculate a landform age of 11,380 ±830

yrs, rounded to **11.4 ±0.8 ka**. **J4R** outside J3R gives a moraine age of 10,750 ±690 yrs (**10.8 ±0.7 ka**) derived from boulder ages of samples JAM-18-04 (10,680 ±200 yrs), JAM-19-21 (10,920 ±210 yrs) and JAM-19-22 (10,660 ±130 yrs) (Fig. 7a–b).

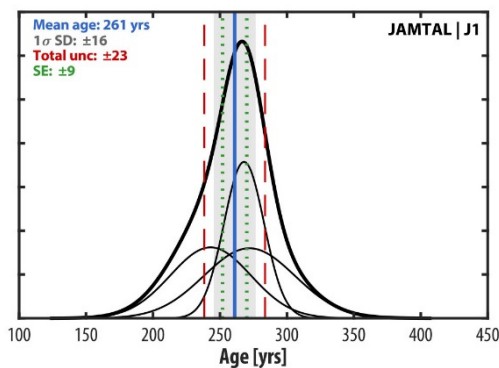

**Figure 6: Kernel plot of LIA ages produced from boulders embedded in the historical moraine at Jamtal. Grey-shaded bars illustrate 1σ landform age uncertainties calculated from analytical uncertainties of individual samples (c. 3 %). Red dashed lines mark**
**uncertainties reported with landform ages including production rate uncertainties (c. 6.3 %).**





**Table 1: [10]Be analytical data and corresponding exposure ages of Jamtal samples. Samples are grouped according to the landforms they were sampled from. Samples were analyzed at the CAMS-LLNL. All samples were measured against the 07KNSTD3110 standard with a ratio of 2.85 x 10[-12] (Nishiizumi et al., 2007). Two procedural blanks were processed with each batch of samples with ratios ranging from 2.8 to 9.1 x 10[-16] (supplements Table S2). The [10]Be background contaminations measured in the blanks were subtracted from the samples. Exposure ages were calculated with the calculator formerly known as CRONUS-Earth online calculator (v3) (Balco et al., 2008), using the Swiss [10]Be production rate (Claude et al., 2014) and choosing the 'Lm' scaling scheme. Ages are calculated relative to the sampling year denoted by the first number in the sample ID and rounded to the nearest 10 years. Uncertainties of boulder ages include the analytical error and a 1 % uncertainty on the carrier concentration.**

| Sample ID | Latitude | Longitude | Elev. | Av. Thickness | Shielding factor | Quartz weight | [9]Be added | [10]Be/[9]Be ratio ±1σ analytical unc. (10[-14]) | | | [10]Be/[9]Be ratio ±1σ analytical unc. | | | [10]Be conc. ±1σ analytical unc. | | | [10]Be exposure age ±1σ analytical unc. and carrier unc. | | |
|---|---|---|---|---|---|---|---|---|---|---|---|---|---|---|---|---|---|---|---|
| | [DD] | [DD] | [m a.s.l.] | [cm] | | [g] | [µg] | | | | [atoms] | | | [atoms/g qtz] | | | [yrs] | | |
| JAM-18-06 | 46.8823 | 10.1684 | 2223 | 1.53 | 0.9317 | 13.3800 | 186 | 0.70 | ± | 0.09 (12.6%) | 86597 | ± | 10881 | 6141 | ± | 772 | **270** | ± | **30** |
| JAM-18-17 | 46.8774 | 10.1723 | 2297 | 1.10 | 0.9760 | 10.4745 | 186 | 0.55 | ± | 0.07 (12.5%) | 67819 | ± | 8457 | 6052 | ± | 755 | **240** | ± | **30** |
| JAM-18-18 | 46.8776 | 10.1725 | 2289 | 2.87 | 0.9748 | 23.4243 | 186 | 1.30 | ± | 0.07 (5.4%) | 161887 | ± | 8777 | 6528 | ± | 354 | **270** | ± | **20** |
| JAM-18-07 | 46.8820 | 10.1682 | 2231 | 1.55 | 0.9044 | 17.4383 | 187 | 4.59 | ± | 0.15 (3.3%) | 572357 | ± | 18941 | 32307 | ± | 1069 | **1500** | ± | **50** |
| JAM-18-16 | 46.8766 | 10.1718 | 2316 | 2.37 | 0.9770 | 31.2824 | 186 | 6.51 | ± | 0.14 (2.1%) | 804960 | ± | 16997 | 25445 | ± | 537 | **1070** | ± | **20** |
| JAM-18-01 | 46.8766 | 10.1736 | 2350 | 1.59 | 0.9557 | 31.1628 | 185 | 67.00 | ± | 1.22 (1.8%) | 8250915 | ± | 149649 | 264480 | ± | 4797 | **11020** | ± | **200** |
| JAM-18-02 | 46.8767 | 10.1735 | 2339 | 1.70 | 0.9752 | 11.4783 | 186 | 25.38 | ± | 0.41 (1.6%) | 3151014 | ± | 51230 | 274134 | ± | 4457 | **11280** | ± | **180** |
| JAM-18-03 | 46.8770 | 10.1734 | 2328 | 4.00 | 0.9759 | 7.2439 | 186 | 16.50 | ± | 0.31 (1.9%) | 2045059 | ± | 38161 | 281704 | ± | 5257 | **11850** | ± | **220** |
| JAM-18-04 | 46.8771 | 10.1739 | 2335 | 2.59 | 0.9748 | 20.2670 | 183 | 42.55 | ± | 0.79 (1.9%) | 5196242 | ± | 96265 | 255946 | ± | 4742 | **10680** | ± | **200** |
| JAM-19-21 | 46.8776 | 10.1738 | 2318 | 3.20 | 0.9756 | 10.4519 | 185 | 21.79 | ± | 0.41 (1.9%) | 2703364 | ± | 50631 | 258051 | ± | 4833 | **10920** | ± | **210** |
| JAM-19-22 | 46.8773 | 10.1738 | 2329 | 1.63 | 0.9729 | 10.6823 | 185 | 22.09 | ± | 0.28 (1.2%) | 2739711 | ± | 34113 | 255888 | ± | 3186 | **10660** | ± | **130** |

### 4.2.2 Laraintal

Sample LAR-19-23 (700 ±20 yrs) stems from **L1** and captures an early LIA maximum early in the 14[th] century (Fig. 4, Table 2). The deposition of the left-lateral **L3L** moraine is constrained by LAR-19-22 with a boulder age of 11,210 ±210 yrs and by LAR-19-24 dated to 10,930 ±210 yrs. Its right-lateral equivalent **L3R** yields boulder ages of 11,120 ±210 yrs (LAR-19-13) and 11,200 ±280 yrs (LAR-19-14). The age of the terminal segment of L3 – L3T – is derived from samples LAR-18-09 (11,540 ±200 yrs), LAR-19-17 (11,070 ±210 yrs), LAR-19-18 (11,330 ±290 yrs), LAR-19-19 (10,730 ±290 yrs), LAR-19-20 (11,480 ±210 yrs) and results in a landform age of 11 230 ±780 yrs. Based on our mapping and dating results, we are confident that all three moraine segments L3L, L3R and L3T can be attributed to the same glacier advance or stabilization dated to. Therefore, we aggregate all nine boulder ages from **L3** and compute a moraine age of 11 180 ±750 yrs (**11.2 ±0.8 ka**). Based on the same reasoning, we combine L4R (LAR-19-12: 10,890 ±180 yrs, LAR-19-15: 10860 ±200 yrs, LAR-19-16: 11060 ±220 yrs – landform age 10 940 ±690 yrs) and L4T (LAR-18-03: 10,160 ±190 yrs, LAR-18-10: 10,880 ±210 yrs, LAR-19-21: 10,660 ±220 yrs – landform age 10 570 ±760 yrs) and suggest a moraine age of 10750 ±740 yrs (**10.8 ±0.7 ka**; Fig. 7c–d).

Analytical results from samples, which were spiked with Fe, show that ages calculated from both samples are consistent with boulder ages obtained for the same landforms, but processed according to the standard protocol (L3R: LAR-19-13, L4R: LAR-



19-12 and LAR-19-15). Analytical uncertainties of corresponding samples amount to 2.0 % (LAR-19-16) and 2.5 % (LAR-19-14) and are within the expected range of $^{10}$Be AMS measurement uncertainties at LLNL-CAMS (Rood et al., 2013). By replacing a portion of the $^9$Be carrier with Fe, we achieved similar analytical precision as with routinely processed samples,
but with only a quarter of the sample mass used. Our results suggest that the substitution of a fraction of $^9$Be carrier using Fe is a viable and promising advancement in the sample preparation protocol that extends the application field of the $^{10}$Be SED method to younger samples and more challenging lithologies.

**Table 2: $^{10}$Be analytical data and corresponding exposure ages of Laraintal samples. For details on analytics, processing or age calculation see captions of Table 1 and supplements Tables S1 and S2. Samples LAR-19-14 and LAR-19-16 were spiked with Fe in addition to (a reduced amount of) $^9$Be carrier.**

| Sample ID | Latitude | Longitude | Elev. | Av. Thickness | Shielding factor | Quartz weight | $^9$Be added | Fe added | $^{10}$Be/$^9$Be ratio ±1σ analytical unc. (10$^{-14}$) | | | $^{10}$Be/$^9$Be ratio ±1σ analytical unc. | | $^{10}$Be conc. ±1σ analytical unc. | | $^{10}$Be exposure age ±1σ analytical unc. and carrier unc. | | |
|---|---|---|---|---|---|---|---|---|---|---|---|---|---|---|---|---|---|---|
| | [DD] | [DD] | [m a.s.l.] | [cm] | | [g] | [µg] | [µg] | | | | [atoms] | | [atoms/g qtz] | | [yrs] | | |
| LAR-19-23 | 46.9234 | 10.2216 | 2292 | 1.48 | 0.9433 | 30.2276 | 181 | | 4.20 ± | 0.10 | (2.3%) | 507590 ± | 11631 | 16570 ± | 380 | 700 | ± | 20 |
| LAR-18-09 | 46.9272 | 10.2225 | 2177 | 2.03 | 0.8899 | 18.9009 | 186 | | 35.05 ± | 0.62 | (1.8%) | 4340818 ± | 76283 | 229187 ± | 4028 | 11540 | ± | 200 |
| LAR-19-17 | 46.9267 | 10.2223 | 2179 | 1.52 | 0.9424 | 10.1141 | 182 | | 19.47 ± | 0.36 | (1.9%) | 2366107 ± | 43845 | 233453 ± | 4326 | 11070 | ± | 210 |
| LAR-19-18 | 46.9268 | 10.2223 | 2178 | 2.24 | 0.9427 | 10.5868 | 182 | | 20.76 ± | 0.54 | (2.6%) | 2520432 ± | 65068 | 237607 ± | 6134 | 11330 | ± | 290 |
| LAR-19-19 | 46.9267 | 10.2226 | 2178 | 1.56 | 0.9413 | 10.2762 | 185 | | 18.72 ± | 0.35 | (1.9%) | 2319925 ± | 42983 | 225277 ± | 4174 | 10730 | ± | 200 |
| LAR-19-20 | 46.9267 | 10.2224 | 2180 | 2.52 | 0.9434 | 10.1923 | 182 | | 20.22 ± | 0.38 | (1.9%) | 2460269 ± | 45624 | 240901 ± | 4467 | 11480 | ± | 210 |
| LAR-19-13 | 46.9224 | 10.2258 | 2303 | 1.26 | 0.9604 | 10.2457 | 180 | | 22.20 ± | 0.41 | (1.9%) | 2676391 ± | 49575 | 260739 ± | 4830 | 11120 | ± | 210 |
| LAR-19-14 | 46.9225 | 10.2258 | 2303 | 1.96 | 0.9584 | 2.6078 | 102 | 99 | 10.02 ± | 0.25 | (2.5%) | 686210 ± | 17191 | 260711 ± | 6531 | 11200 | ± | 280 |
| LAR-19-22 | 46.9245 | 10.2213 | 2256 | 1.89 | 0.8546 | 10.3313 | 181 | | 19.30 ± | 0.36 | (1.9%) | 2335406 ± | 43280 | 225574 ± | 4180 | 11210 | ± | 210 |
| LAR-19-24 | 46.9225 | 10.2201 | 2357 | 1.67 | 0.9581 | 10.4515 | 185 | | 22.38 ± | 0.42 | (1.9%) | 2767560 ± | 51798 | 264203 ± | 4945 | 10930 | ± | 210 |
| LAR-18-03 | 46.9273 | 10.2220 | 2162 | 1.54 | 0.9245 | 24.1911 | 186 | | 40.30 ± | 0.75 | (1.9%) | 5004558 ± | 92711 | 206505 ± | 3826 | 10160 | ± | 190 |
| LAR-18-10 | 46.9272 | 10.2225 | 2161 | 2.77 | 0.9401 | 6.2606 | 187 | | 11.25 ± | 0.21 | (1.9%) | 1404082 ± | 26340 | 223566 ± | 4194 | 10880 | ± | 210 |
| LAR-19-21 | 46.9273 | 10.2222 | 2160 | 2.25 | 0.9383 | 10.3970 | 180 | | 18.92 ± | 0.35 | (1.9%) | 2281226 ± | 42291 | 218937 ± | 4059 | 10660 | ± | 200 |
| LAR-19-12 | 46.9231 | 10.2261 | 2295 | 1.36 | 0.9585 | 10.0087 | 182 | | 20.82 ± | 0.33 | (1.6%) | 2536338 ± | 40705 | 252920 ± | 4059 | 10890 | ± | 180 |
| LAR-19-15 | 46.9240 | 10.2259 | 2275 | 3.47 | 0.9563 | 10.4801 | 182 | | 21.05 ± | 0.39 | (1.9%) | 2561640 ± | 47473 | 243958 ± | 4521 | 10860 | ± | 200 |
| LAR-19-16 | 46.9238 | 10.2260 | 2280 | 3.18 | 0.9453 | 2.5099 | 104 | 99 | 9.01 ± | 0.18 | (2.0%) | 627353 ± | 12552 | 247430 ± | 4950 | 11060 | ± | 220 |





**Figure 7: Kernel plots of Holocene moraine ages. Left column (a) and (c): inner dated EH moraines at Jamtal and Laraintal. Right column (b) and (d): outermost dated EH moraines at Jamtal and Laraintal. (e) Synthesis of moraine ages across both valleys grouped to Moraine Formation Intervals (MFI). MFI 3 (calculated from J3R and L3) yielding an age of 11,230 ±760 yrs rounded to 11.2 ±0.8 ka. (f) MFI 4 (calculated from J4R and L4) yielding an age of 10,750 ±720 yrs rounded to 10.8 ±0.7 ka. Grey-shaded bars illustrate 1σ landform age uncertainties calculated from analytical uncertainties of individual samples (c. 3 %). Red dashed lines mark uncertainties reported with landform ages including production rate uncertainties (c. 6.3 %), which results in conservative uncertainty estimates.**



## 5 Discussion

### 5. 1 The moraine record of the past two millennia

The classical LIA moraines of both valleys (J1 and L1) feature boulders deposited within the expected time interval, i.e.
between 1250 and 1850 CE (Fig. 8f). An advance of Larainferner to the LIA moraine is recorded at the beginning of 14th century (LAR-19-23) and suggests an early LIA culmination in this valley. Three consistent boulder ages from J1 are aggregated to a mean age of 260 ±25 yrs and indicate an advance of Jamtalferner between c. 1735 and 1790 CE. A recent geochronological study in the adjacent Ochsental comes to remarkably similar results with boulder ages from the LIA moraine yielding a mean age of 260 ±30 yrs (Fig. 8e) (Braumann et al., 2020). Glacier advances during this period are also documented
in the Eastern Alps, for instance at the Zillertal and at the Ötztal (Pindur and Heuberger, 2010;Nicolussi, 2013), in the Central Alps at the Lower Grindelwald glacier (Zumbühl and Nussbaumer, 2018) and in the Western Alps at the Mer de Glace (Nussbaumer et al., 2007). High glacial activity during the second half of the 18th century with termini coming close to, or reaching their LIA maximum, is congruent with a phase of decreased summer temperatures detected in proxy records in the vicinity of our study site (Fig. 8a–b) (Ilyashuk et al., 2019;Larocque-Tobler et al., 2010a;Fohlmeister et al., 2013;Vollweiler
et al., 2006), and with reconstructed summer and mean annual temperatures from Greenland ice cores (Fig. 8d) (Kobashi et al., 2017;Buizert et al., 2018).

Besides LIA-aged boulders along the LIA moraine we sampled two blocks of J1, which were deposited during the first millennium of the common era. The younger boulder, JAM-18-16, dates to the beginning of the Medieval Warm Period (MWP). By that time, glaciers in the region were likely smaller relative to their LIA maxima (e.g. Solomina et al., 2016). The
boulder's position and its bedding were re-evaluated in the field after age calculation, and we cannot exclude that the boulder has tilted (Fig. S7). Therefore, we interpret the exposure age as a minimum age. The older neoglacial boulder, JAM-18-07, was exposed 1500 ±50 yrs ago, which is again in good agreement with the neighboring Ochsental chronology, where a block in an identical setting (embedded in the classical LIA moraine) – was dated to 1500 ±40 yrs (Fig. 8e) (Braumann et al., 2020). Evidence for a period of glaciers advance in the Eastern Alps during the 5th and 6th century CE was found beyond the Silvretta
region, for instance in sediment profiles and peat cores in the forefield of Fernauferner, Mittelbergferner and Simonykees (Patzelt, 2016;Patzelt and Bortenschlager, 1973). This timing coincides with prominent episodes of glacier advance in the Western Alps, most notably at the Aletsch glacier (Holzhauser et al., 2005), and at the Miage and Mer de Glace, both in the Mont Blanc massif (Deline and Orombelli, 2005;Le Roy et al., 2015). Concurrent glacier advances have also been reported from Alaska, Iceland, Scandinavia and from Greenland (Biette et al., 2020;Barclay et al., 2009;Solomina et al., 2016). Glacier
advance during this period is consistent with decreasing summer temperatures (Fig. 8c–d) and higher precipitation rates in Europe (e.g. Büntgen et al., 2011). This regional climate perturbation, which is often referred to as Dark Ages Cold Period (DACP) and which occurred in tandem with the migration period in Europe, began around 400 CE and lasted into the 8th century CE in the region (e.g. Helama et al., 2017). During that time, the Atlantic Meridional Overturing Circulation (AMOC), which transports heat from the South Atlantic towards the North was weakened (Thornalley et al., 2018), which lead to cooling





in the Nordic region. Potential volcanic eruption(s) in the Northern hemisphere in the year of 536 CE may have amplified

cooling across Europe and define the onset of the recently postulated Late Antique Little Ice Age (LALIA; 536 to c. 660 CE)

(Büntgen et al., 2016). Consistent with the timing of the regional DACP, Helama et al. (2021) suggests centennial scale phases

in Northern Europe during the Holocene, which resemble the LIA climatic regime, one among them beginning around 540

CE.

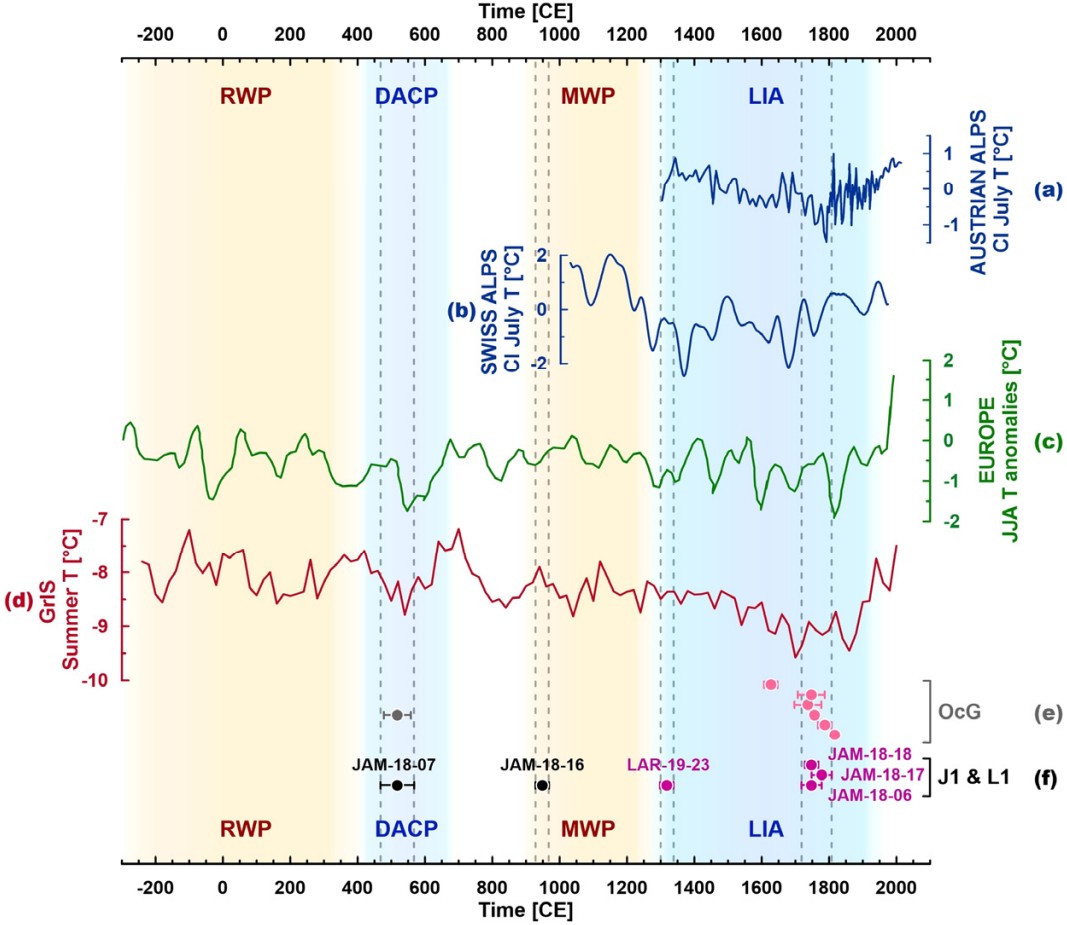


**Figure 8: Youngest part of the ¹⁰Be chronology from Jamtal and Laraintal correlated with climate proxy data from the Alps (local records in dark blue), Europe (green) and the Greenland Ice Sheet (GrIS) covering the past c. 2000 yrs. Proxy records indicate cooler climate conditions during the Dark Ages Cold Period (DACP) and during the Little Ice Age (LIA), synchronous with periods of moraine formation in the Silvretta region. Chironomid-inferred (CI) July temperatures from (a) Mutterbergersee in the Stubai**
**Alps, Austria (Ilyashuk et al., 2019) c. 70 km E of study site (north of the Alpine drainage divide) and (b) lake Silvaplana c. 60 km SW of study site (Engadin, Switzerland; south of the Alpine drainage divide) (Larocque-Tobler et al., 2010a). (c) European summer temperature anomalies (reference period 1961-1990, 60-year low-pass filter) identified in tree-ring chronologies (Büntgen et al., 2011). (d) Mean summer temperature reconstructions (JJA) derived from nine Greenland ice cores (Buizert et al., 2018). (e) ¹⁰Be ages of boulders sampled from the presumable LIA (Holocene composite moraine) from Ochsental (OcG) (Braumann et al., 2020)**
**and from (f) Jamtal and Laraintal (this study). RWP – Roman Warm Period; MWP – Medieval Warm Period.**



In summary, samples collected along the classical LIA moraine at the Jamtal (this study) and at the adjacent Ochsental (Braumann et al., 2020) yield ages that fall into the regional DACP and the LIA. These results are consistent with the timing of glacier advances across the Alps and in other places of the Northern hemisphere. The advance of Silvretta glaciers coincides
with cooling trends captured in local, regional and hemispheric proxy data. Moraines J1 and L1 are composite moraines that were at least reached once prior to the LIA. J1 and L1, in the following termed 'Holocene composite moraines', mark the amplitude of glacier advances and temperature minima during Holocene interglacial, when the YD-EH transition was concluded.

**5. 2 The moraine record of the past two millennia**

**5.2.1 Local correlation**

Moraine records at Jamtal and at Laraintal are remarkably similar and point to synchronous glacier dynamics throughout the Holocene, particularly during its onset. In both valleys, we identified up to three lateral ridges just outboard the J1 and L1 composite moraines, and their terminal equivalents, albeit in varying states of preservation. The outermost ridges in both valleys, J4R and L4 yield identical landform ages (Fig. 7b and 7d). As we attribute these landform ages to a climatic state, we
group them across both valleys and refer to them as **Moraine Formation Interval (MFI) 4,** equivalent to **10.8 ±0.7 ka** (Fig. 7f). We proceed in the same way with moraines J3R and L3 (Fig. 7a and 7c) and aggregate their ages to a second group termed **MFI 3: 11.2 ±0.8 yrs** (Fig. 7e). MFI 3 and 4 overlap within 1σ uncertainties. Field evidence of a multi-ridge structure implies two phases of glacier advances or stabilization in the region.

We correlate EH Jamtal and Laraintal moraine chronologies with moraine chronologies and glacier proxy records at the local
scale and propose a concept of YD-EH deglaciation. We note that climate (variability) is roughly constant across the valleys in Figure 1b. Also, catchments are by and large comparable in terms of elevation, glacier size, exposure, geographical location (North of the Alpine drainage divide), and orientation (S–N). Hence, glaciers in the region respond(ed) to the same climate forcings, and probably at a similar level of sensitivity. Variations in the timing of moraine formation point to distinct episodes of high glacier activity, which have superimposed the general warming trend during the YD-EH transition. Other explanations
for age variability among moraines dated in the region could be uncertainties tied to the dating method, or catchment-specific effects such as shading or bedrock topography. However, local effects are perhaps minor in comparison to glacier responses to climate forcing, particularly in the light of the vast temperature increase during the YD-EH transition.

MFI 3 and 4 both fall well into the EH and are different from advances during the LG. Presumable YD moraines are identified at considerable distance downstream and outboard of landforms addressed in this study (Hertl, 2001, and references therein),
which implicates that glaciers shrank from their LG ice margin to a position close to the LIA maximum within a few centuries. Rapid deglaciation is a direct response of glaciers to an increase of summer temperatures by several degrees during the YD-EH transition, in the Eastern Alpine region and across the Alps (9c-f) (e.g. Ilyashuk et al., 2009;Heiri et al., 2014;Samartin et al., 2012;Larocque-Tobler et al., 2010b;Affolter et al., 2019). This warming trend was interrupted by brief cold spells, which



manifest in moraine records in the Silvretta Massif and in the adjacent Verwall mountains. Based on these moraine
chronologies, we suggest the following local YD-EH glacier history illustrated in Figure 10:

i. **Late YD — Kartell moraines (Verwall):** Moraines in the region that formed within this period have been dated at the
Kartell site, c. 15-20 km NE of Jamtal and Laraintal (Ivy-Ochs et al., 2006). Because of the previously higher $^{10}$Be
production rate estimate, Kartell moraines were initially placed into the EH. Recalculations using the updated production
rate yield boulder ages ranging between 11.8 ±0.6 ka and 12.5 ±0.9 ka with the production rate uncertainty excluded. The
age update now assigns Kartell moraines to the late(st) YD (Ivy-Ochs, 2015;Boxleitner et al., 2019b). A local lowering
of the ELA of approximately 120 m relative to the historical moraine was reported by the authors.

ii. **YD-EH transition/Preboreal — MFI 3 and MFI 4:** The shift from glacial to interglacial conditions is captured at the
Jamtal and Laraintal. Corresponding moraine chronologies indicate ice margins at terminal positions some hundreds of
meters outboard of the LIA maximum, equivalent to an estimated ELA depression of c. 70 m (Hertl, 2001:80). These
chronologies evidence abrupt cold snaps, which interrupted the general warming trend during the EH and which caused
decadal to centennial scale glacier oscillations. The mean age of MFI 3 (11.2 ka ±0.8) falls within **the Preboreal
Oscillation (PBO)** as defined based on paleoenvironmental records from Europe (11.30–11.15 ka) (e.g. Bjorck et al.,
1997;Schwander et al., 2000;Magny et al., 2007;Joannin et al., 2013)**;** MFI 4 with a central age of 10.8 ±0.7 ka postdates
the PBO but correlates with summer cooling detected in lake sediments in the Swiss and Austrian Alps (Fig. 9c–d) (Heiri
et al., 2003;Lauterbach et al., 2011).

Kromer moraines identified in valleys 10-15 km further towards the West (Fig. 1b) were originally placed into the
Preboreal (Gross et al., 1978). Morphologically, these moraines resemble the blocky, multi-ridge structures of J3/L3 and
J4/l4 at the Jamtal and Laraintal and yield similar snowline depression estimates of c. 70–90 m. Updated $^{10}$Be moraine
ages of 9.9 ±0.7 ka and 10.2 ±0.7 ka fall within a somewhat younger age spectrum compared to the Jamtal and Laraintal
moraines (Kerschner et al., 2006;Moran et al., 2016b). However, the age discrepancy between Kromer moraines on the
one hand and Jamtal and Laraintal moraines on the other hand could be reconciled considering age uncertainties.

iii. **Interglacial/Holocene mode — Ochsental–Grüne Kuppe:** Deglaciation patterns in the Ochsental to the West suggest
ice margin configurations similar to the LIA around 10 ka. The timing of moraine formation adjacent to the lateral
Holocene composite moraine (equivalent to J1 and L1 in this study) was constrained to 9.9 ±0.7 ka (Grüne Kuppe site,
n=4) (Braumann et al., 2020). In addition, two boulders of the same age were found in latero-frontal sections of the
Holocene composite moraine and point to similar ice margins around 10 ka and during the LIA. Although no exposure
ages are available for J2 and L2, and although these moraines could have been deposited within any cold phase between
MFI 3 and the onset of the LIA, we note that J2 and L2 potentially may correlate with the Grüne Kuppe moraine,



particularly as the ridges are in a morphostratigraphically similar position. The timing of the Grüne Kuppe moraine stabilization aligns with a climate anomaly detected in some proxy records of the Alps around 10.5 ka which persisted for several centuries, sometimes termed **Central European cold phase 1** (CE-1) (e.g. Haas et al., 1998;Schmidt et al., 2006;Boch et al., 2009). This phase was followed by deglaciation and glaciers in the Alps receded to sizes smaller than their historical maximum (Patzelt, 2019;Solomina et al., 2015). Glacier retreat lead to a vegetation change with trees

spreading to high(er) elevations. At the Las Gondas bog in the adjacent Fimbatal (Fig. 1b), subfossil wood and tree logs were found up to an elevation of 2370 m a.s.l., with the oldest sample dated to 8620-8480 cal BP at 2355 m.a.s.l. (Nicolussi, 2010). In close vicinity to our study sites (Futschöltal), evidence of pinus cembra populations evolving at an elevation of c. 2290 m a.s.l. within the period between c. 5580 and 4970 calBP was found (Patzelt, 2019). Consistent results were reported from other valleys in the region, for instance from the Klostertal and the Bielerhöhe sites (Nicolussi, 2010), and

from Kaunertal (Nicolussi et al., 2005) (Fig. 1b).



**Figure 9: Proxy records capturing the YD-EH transition. (a)** Detrical Carbonate Peaks (DCP) identified in marine sediment cores from the Labrador sea (Jennings et al., 2015); TC – Total Carbonate. **(b)** Mean summer temperature reconstructions (JJA) derived from nine Greenland ice cores (Buizert et al., 2018). **(c)** Ostracods record extracted from lake sediments of Mondsee (Austrian Alps) (Lauterbach et al., 2011). **(d)** Chironomid-inferred (CI) atmospheric July temperatures from lake sediments of Hinterburgsee in the Swiss Alps. **(e)** Stacked CI July temperatures in the European Alps (Heiri et al., 2014). **(f)** Mean Annual Temperatures (MAT) in Central Europe reconstructed based on speleothems from Milandre Cave, Switzerland (Affolter et al., 2019). **(g)** Arctic moraine record: GrIS – Greenland Ice Sheet moraines, LIS – Laurentide Ice Sheet (LIS) moraines, NA Alpine – North America Alpine mountain glacier moraines (Young et al., 2020). **(h)** Moraine Formation Intervals (MFI) identified in this study. Purple bars highlight Holocene cold events detected in Greenland ice cores (Rasmussen et al., 2007).





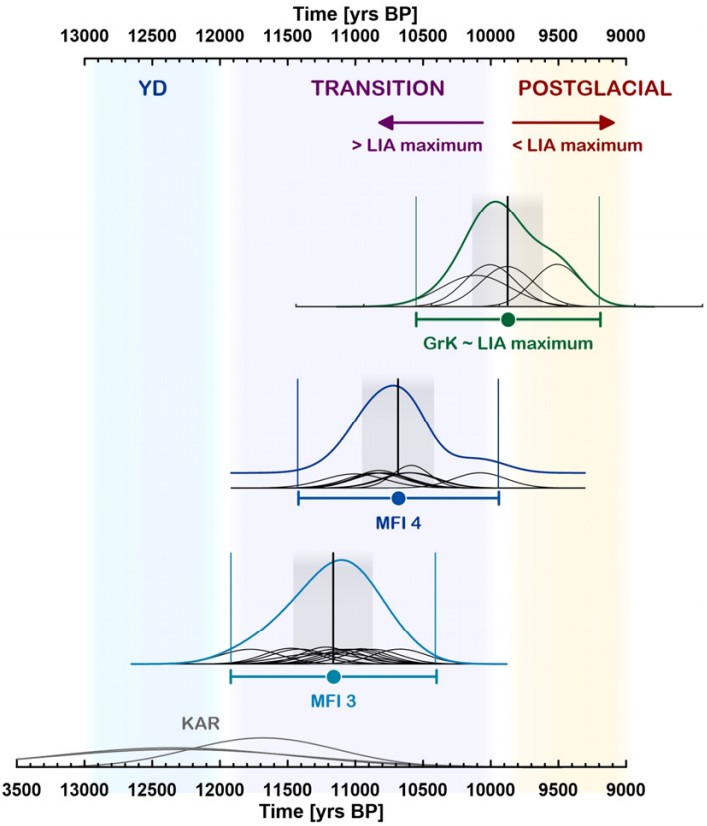

**Figure 10: Glacier retreat during the transition from glacial to interglacial conditions evidenced in moraine chronologies from the Silvretta and Verwall region. KAR – Kartell moraines (Ivy-Ochs et al., 2006;Ivy-Ochs, 2015), site is located c. 15-20 km NE of Jamtal**
**and Laraintal. MFI 3 and 4 described in this study. GrK – Grüne Kuppe moraine identified in the adjacent Ochsental suggesting LIA-like glacier extents around 10,000 years (Braumann et al., 2020).**

### 5.2.2 Alpine-wide and hemispheric correlation

Moraine formation during the transition from glacial to interglacial climatic conditions that is presented in Figure 10, builds

on glacier records in the Silvretta Massif and in the Verwall mountains in the Eastern Alps. This model is consistent with

results of previous studies that have addressed glacier evolution during LG and EH based on moraine chronologies (e.g. Protin

et al., 2019;Schindelwig et al., 2012;Schimmelpfennig et al., 2012;Schimmelpfennig et al., 2014;Moran et al., 2016a;Moran

et al., 2017a;Hofmann et al., 2019;Baroni et al., 2017;Protin et al., 2021). Investigated moraine sets may differ with respect to

their structure, state of preservation or distance relative to the LIA maximum, but they share their position (outboard the LIA

maximum but inboard the presumable LG ice margin), and their age of deposition between c. 12 and 10 ka before present and

imply significant large scale cooling during this period. This pattern of EH moraine stabilization is not limited to the Alpine

realm, but has been observed in other places in the North Atlantic and Arctic region, for instance along the Fennoscandian ice





sheet (e.g. Briner et al., 2014;Nesje, 2009), the Icelandic ice sheet (e.g. Sigfusdottir and Benediktsson, 2020), at Svalbard (e.g. Farnsworth et al., 2020), along the eastern part of the LIS (e.g. Young et al., 2020;Corbett et al., 2016;Ullman et al., 2016) and

along the Greenland ice sheet (e.g. Young et al., 2020;Biette et al., 2020;Levy et al., 2016) (Fig. 9g). Atmospheric temperatures were certainly different in these regions during the EH (Fig. 9b–f), and glaciers in the European Alps retreated much earlier to positions inboard their subsequent historical margin compared to Arctic glaciers which continued to deposit moraines outboard their LIA at least two more millennia. However, concurrent moraine stabilization during the EH raises the question what caused this synchronicity in climatic cooling during the first millennia of the Holocene.


### 5.2.3 Climatic drivers of EH moraine formation in the Northern hemisphere

Freshwater influx into the North Atlantic and into the Arctic Ocean is known as a driver for climate of the northern hemisphere and acts as a plausible cause for abrupt centennial scale cold snaps during the LG and the EH (e.g. Bjorck et al., 1997;Nesje et al., 2004;Fisher et al., 2002;Hald and Hagen, 1998;Thornalley et al., 2010). Prominent examples are repeated outbursts of

the north American proglacial Agassiz lake, whose final drainage caused a sharp temperature drop in the northern hemisphere, the 8.2 ka event detected in Greenland ice cores (Hillaire-Marcel et al., 2007;Thornalley et al., 2010;Alley and Agustsdottir, 2005;Clarke et al., 2009;Teller et al., 2002). Besides abrupt high-volume releases of freshwater to the North Atlantic or Arctic Ocean via major lake drainages or iceberg armadas, there is evidence of more subdued glacial discharge during the EH that results in a deceleration of the thermohaline circulation. Weakening of the AMOC leads to less heat transported to the North

Atlantic region, which can prompt brief, decadal to centennial scale cold snaps in the North, and also at lower latitudes. During the PBO and subsequent centennial scale cold phases, harsher climate conditions are reported in the North Atlantic region (e.g. Knudsen et al., 2008;Timms et al., 2021;Paus et al., 2015;Bos et al., 2007) with cooling extending towards Western and Central Europe (Fig. 9c-f). Glaciers in glaciated North Atlantic regions and in the Alps responded to these climate perturbations with stabilization or advance, hence moraine deposition.

To test the linkage between EH moraine formation and freshwater discharge of the LIS, we review corresponding markers in marine sediment cores, temperature proxy records and moraine records in these regions. We begin with the YD termination, when ice bergs and meltwater plumes were released into the North Atlantic, evidenced by ice-rafted debris and layers of 'foreign' sediment enriched with detrital carbonates in marine sediments. These layers, often referred to as Heinrich-0 (H0) and characterized by a Detrical Carbonate Peaks (DCP) date to the earliest Holocene and were identified in Baffin Bay (Simon

et al., 2014), in the Labrador Sea (Rashid et al., 2011;Andrews et al., 1995), at the coast of Newfoundland (Pearce et al., 2015), including the Flemish cap (Li and Piper, 2015). Jennings et al. (2015) found eight subsequent detrital carbonate peaks (DCP 1–6) between 11,500 and 8,000 yrs BP in a core from the Labrador shelf, which is attributed to freshwater sourced from Hudson Strait (Fig. 8a). **DCP1** detected around 11,500 yrs BP coincides with the end of H0 and with the onset of 11.4 ka event as (c. 11,450 to 11,350 cal BP) (Rasmussen et al., 2007). During the subsequent PBO captured in Nordic records, mean annual

and summer temperatures in the European Alps declined (Fig. 8e–f) (e.g. Affolter et al., 2019;Lauterbach et al., 2011;Heiri et





al., 2014). The propagation of this cooling trend towards Western and Central Europe is supported by cooler and more humid climate conditions in these regions c. 11,300–11,150 cal BP (Magny et al., 2007). In parallel, [10]Be concentration in Greenland ice cores, a proxy for solar activity, decreases towards a minimum (Finkel and Nishiizumi, 1997;Mekhaldi et al., 2020;Adolphi et al., 2014). Low solar activity may have amplified the cooling imposed by freshening of the Atlantic Ocean, or vice versa.

Glaciers in the North Atlantic region and in the European Alps advanced or stabilized repeatedly during the first millennia of the Holocene and deposited moraines, evidenced by moraines J4 and L4 dated in this study (Fig. g–h).

A similar chain of events may have occurred some centuries later. The deposition of the **DCP2** c. 10,600 cal BP was preceded by the so-called Gold Cove advance of the LIS's Labrador sector across Hudson Strait and its subsequent retreat (Rashid et al., 2014;Jennings et al., 2015;Kaufman et al., 1993). Resulting freshwater input may have weakened the AMOC, which in

turn lead to a drop in mean annual temperatures in Greenland and moraine formation in the Arctic (e.g. Young et al., 2020;Biette et al., 2020). Temperatures in the Alps decreased or stagnated around that time (Fig. 9c–d, f). Moraine formation in the Alps between 10,700 yrs BP and 10,500 yrs BP, concurrent with DCP2, is observed in the Silvretta Massif (MIF 4) and across the European Alps (Schimmelpfennig et al., 2014;Protin et al., 2019;Schimmelpfennig et al., 2012;Moran et al., 2017a). The linkage between DCPs, freshwater input into the Atlantic and Arctic Ocean and subsequent EH glacier advances has been

put forward before in the context of the North American and Arctic region (e.g. Young et al., 2020;Andrews et al., 2014;Nesje, 2009). The authors of a recent geochronological study carried out in the Western Alps go a step further and propose that freshwater forcing in the North Atlantic region acted as a driver for moraine formation in the Mont Blanc Massif (Protin et al., 2021). They suggest that a decrease in AMOC strength lead to extended sea ice periods during winter in the North Atlantic, which in turn caused a southwards shift of the westerlies. Cold air was then transported to Europe and lead to moraine formation

in the European Alps. With our new moraine chronologies from the Silvretta, we complement the glacier record from the Western Alps with robust evidence for EH moraine formation in the Eastern Alps and corroborate the hypothesis that centennial scale cold phases occurred at a regional (hemispheric) scale between 12 and 10 ka.

## 6 Summary and Conclusions

    ▪  Glaciers at both study sites, the Jamtal and the Laraintal, stabilized or advanced at least twice during the EH, which is

evidenced by moraines deposited outboard the historical moraine (LIA maximum) and inboard the presumable LG ice margin. The timing of moraine formation is consistent across both valleys and is constrained with [10]Be SED, yielding ages of 10.8 ±0.7 ka (MFI 4, n=9) and 11.2 ±0.8 ka (MFI 3, n=12). EH moraines in the Silvretta region indicate repeated punctuation of the general postglacial warming trend by short cold episodes in Europe. These cold snaps, most prominently the PBO, appear to have their origin in the North Atlantic region. Layers of ice-rafted debris and DCPs in marine sediment

cores along the western margin of the LIS point to glacial discharge during the earliest Holocene. Resulting freshwater released into the North Atlantic probably caused a drop in salinity and led to a weaking of the AMOC. Perturbated heat transport northwards caused cooling in the North Atlantic region, which propagated toward Europe. Glaciers and ice sheets



in the Northern hemisphere responded to this cooling via moraine deposition. We tentatively suggest a similar line of arguments for an episode of glacier stabilization a few centuries later during the EH, c. 10.7–10.5 ka.

▪ Based on Holocene moraine chronologies of the Silvretta Massif and the adjacent Verwall mountains, we propose the following local model that describes alternating phases of glacier retreat and stabilization between 12 and 10 ka: the YD termination (Kartell), (Ivy-Ochs et al., 2006;Ivy-Ochs, 2015), the YD-EH transition (MFI 3 and MFI 4, this study), and the Holocene mode (GrK) (Braumann et al., 2020). The proposed concept confirms the hypothesis formulated by Patzelt and Bortenschlager (1973) almost 50 years ago, that glaciers in the Eastern Alps deposited moraines during the Preboreal, and

had retreated to their subsequent historical ice margins by c. 10 to 9.5 ka. During the rest of the Holocene, the magnitude of cooling was most likely too small in the Eastern Alps to force advances which exceeded dimensions that glaciers had around 10 ka.

▪ Our data suggests that glaciers in the Silvretta region advanced to a position close or equivalent to their LIA maximum around 500 CE. The timing of this advance is concurrent with the migration period in Europe, which is often associated

with regional climate deterioration. There is growing evidence that many glaciers in the Alps, North America and the Nordic region advanced at that time and reached their historical margin (e.g. Holzhauser et al., 2005;Barclay et al., 2009;Patzelt, 2016;Le Roy et al., 2015).

▪ Silvretta glaciers have reached their historical maximum early during the LIA, around 1300 CE. A subsequent advance to the same position took place in the second half of the 18[th] century, documented by three boulders along the historic moraine

yielding a mean age of c. 260 ±25 yrs. This result agrees well with an advance of glaciers in the adjacent Ochsental (Braumann et al., 2020). Contemporaneous advances have also been reported for other glaciated areas beyond the Silvretta region, e.g. Lower Grindelwald glacier, Mer de Glace and for glaciers in the Ötztal (Pindur and Heuberger, 2010;Nicolussi, 2013;Zumbühl and Nussbaumer, 2018;Nussbaumer et al., 2007).

▪ The classical LIA moraine, traditionally referred to as the '1850 moraine' in the European Alps, marks the LIA maximum

glacier extent that was reached multiple times during the LIA, but also earlier during the Holocene, most likely around 500 CE. As the moraine comprises glacial sediments deposited during several glacier advances during the past millennia, we propose that these landforms should rather be viewed as 'Holocene composite moraines' instead of '1850 moraines' or 'LIA moraines'.



**Appendix A: Complementary photo documentation of landforms**

**A1: Jamtal.** (a) J1 moraine with Jamtal hut in the background. Boulder surfaces are fresh and are not colonized by lichens; pioneer plants growing on the moraine. (b) J1 multi-ridge structure with view towards with Jamtalferner in the back. (c) Left-lateral side of Jamtal below Totenfeld glacier. (d) View towards Chalausferner (in the background); J1 in the foreground, J3R and J4R outboard

J1 are curved lateral moraines of former Chalausferner. (e) J2 moraine (undated) at the center with till covered slope to the left and J1 to the right. (f) Right-lateral moraine sets. J4R and J3R date to the Early Holocene, J1 comprises debris that was deposited during the LIA and during at least one earlier glacier advance around 500 CE.





**A2: Jamtal. (a) J5 ridge; view from right-lateral valley side towards W, and (b) view from downvalley position southwards. The**
**structure was reworked in the course of trail construction and maintenance. (c) Transition zone between J1 terminal moraine and**
**blockfield. We speculate that the blockfield originates from rock failures along valley flanks farther upvalley. Corresponding debris**
**was transported downstream by the glacier. During glacier retreat, the blocks melted out and covered the valley floor. The blockfield**
**(at least its uppermost section) was then partly overprinted by subsequent glacier advances. (d) Blockfield consisting of coarse,**
**angular to subangular components; view towards SE. (e) Transition zone between fresh J1 moraine deposits and the blockfield,**
**whose blocks are populated by lichens and are partly overgrown with vegetation. (f) Displaced moraines along the right-lateral**
**valley side outboard J1; Futschöltal (tributary valley) in the back.**



**A3: Laraintal. (a)** Terminal moraines L3T and L4T dissected by a creek (Larainbach). **(b)** Right-lateral moraine set: L1, L2 (undated) and L3R. **(c)** Larainbach, modern river plane with L1 double-ridge structure identified on both sides of the creek. **(d)** Left-lateral flatter valley section, where moraines L3L and L1, and presumable Late Glacial moraines accumulated. **(e)** Left-lateral glacier side. Person standing on L1 ridge with L3L to the left. Hoher Kogel peak in the background. **(f)** Provenance area of rockfall event in 2019, Hoher Kogel peak.





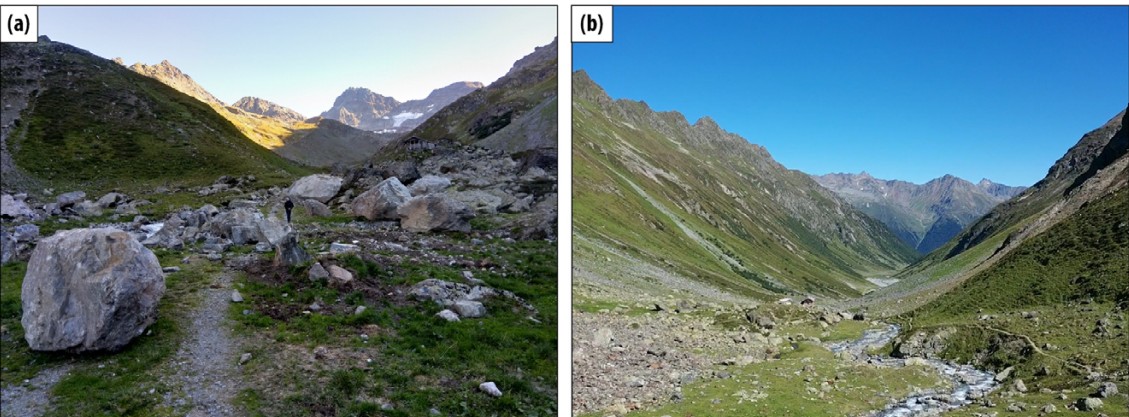


**A4: Laraintal. (a) Fresh rockfall deposits (2019) in the Zollhütte area. Note person at the center of the photograph for scale. (b) Zollhütte area (L5); unweatherd scree on the left side of the photograph; mass movement slab to the right of the hiking trail; fresh rockfall path behind the slab.**

**Data availability**

All analytical information associated with cosmogenic nuclide measurements is listed in the tables in the Supplement and will be made available via the ICE-D Alpine database (http://alpine.ice-d.org/).

**Author contribution**

SMB and JMS designed the study. SMB, MF and JMS carried out field work. SMB was responsible for cosmogenic nuclide sample preparation. AJH performed sample measurements and developed the strategy for low-level $^{10}$Be sample processing. All authors were involved in data interpretation. SMB wrote the manuscript, which was revised and edited by all authors.

**Competing interests**

The authors declare that they have no conflict of interest.

**Acknowledgements**

SMB is grateful to the LDEO cosmogenic nuclide laboratory staff Roseanne Schwartz and Jean Hanley for help and advice during sample processing. Furthermore, SMB thanks Günther Gross for helpful discussions on the Holocene moraine record in the Silvretta Massif, and Dominik Blasenbauer for assistance during field work. The authors express their gratitude to their hosts at the Jamtal hut, at Larainalpe and at Pension Türtscher, who have facilitated the field campaigns. This work was performed in part under the auspices of the U.S. Department of Energy by Lawrence Livermore National Laboratory under Contract DE-AC52-07NA27344; this is LLNL-JRNL-821951.

**Financial support**

This study was sponsored by inatura Museum GmbH. SMB is a recipient of a DOC Fellowship of the Austrian Academy of Sciences (OeAW) at the Institute of Applied Geology, University of Natural Resources and Life Sciences (BOKU) Vienna. For research visits at the LDEO, SMB received a Marietta Blau scholarship sponsored by OeAD GmbH, a Marshall Plan scholarship provided by the Austrian Marshall Plan Foundation and financial support from BOKU's 'Transitions to Sustainability' (T2S) doctoral school.



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
