# Peer review of "Early Holocene cold snaps and their expression in the moraine record of the Eastern European Alps"

_Climate of the Past, 2021_

## Author Comment (AC1)

**Title:**       **Early Holocene cold snaps and their expression in the moraine record of the Eastern European Alps**

**Authors:**   **Sandra M. Braumann, Joerg M. Schaefer, Stephanie M. Neuhuber, Christopher Lüthgens, Alan J. Hidy, Markus Fiebig**

**Article ID:**   **cp-2021-54**

The authors thank Reviewer #1 (R1) very much for reviewing the manuscript and for his/her/their helpful comments. By including R1's suggestions, we believe that the quality of our manuscript has substantially improved. In this document we address in detail all of R1's comments and describe, point by point, the changes in the manuscript. R1's comments are listed in the gray boxes followed by our responses in blue. Line numbers that refer to the changes in the revised manuscript version are listed in red.

**1 | MAIN COMMENTS**

*This manuscript reports new ¹⁰Be exposure ages from Holocene moraines in the Silvretta Mountains in the Eastern European Alps and puts them into the regional and hemispheric paleoclimatic context. The data is very useful, as Holocene chronologies in the Eastern part of the Alps are still scarce, especially such robust, consistent data sets that allow for meaningful paleoclimatic interpretation. The multi-boulder ages are indeed remarkably consistent for each landform and belong thus to the most valuable data sets from the Alps, which are still relatively rare among the many existing Alpine chronologies.*

*The manuscript is very well written and illustrated. Also, a new and very promising method is presented that allows low-quartz or low-¹⁰Be samples to be analyzed with high precision.*

*The manuscript is thus worth to be published in Climate of the Past. Before acceptance, a few minor issues should be addressed though.*

*My main comment relates to the ¹⁰Be mean ages of the Early Holocene (EH) moraine sets that are reversed with regard to the stratigraphic order of the landforms. Two EH ice margins were dated in each of the studied valleys. The multiple ages from each ice margin are strikingly consistent with each other, and their mean ages are in perfect agreement between both valleys. But the ¹⁰Be ages from the outer (thus older) ice margin are systematically younger, thus leading to a nominal mean age that is younger by ~500 years, in both valleys. This is not discussed nor even mentioned in the manuscript. While from a dating perspective this can be handled relatively easily (see my suggestions below), the major challenge consists in the correlation of the moraine depositions with the independent high-resolution proxies (Discussion). This is particularly notable in section 5.3.2, where the cold spells during the EH warming are discussed, and the timing of the outer moraine formation (MIF 4) is correlated with both the DCP1 (+PBO; ~11.6-11.2 ka) and DCP2 (+related cold spell; ~10.6-10.5 ka). See line 531 for the first ("...evidenced by moraines J4 and L4 dated in this study...", i.e. MIF 4), and lines 536-537 for the second ("Moraine formation..., concurrent with DCP2, is observed in the Silvretta Massif (MIF 4)"). Maybe there's a typo and the authors meant MIF 3 for the first, as the 10Be ages MIF 3 agree with DCP1. But that would of course make no sense, because a stratigraphically older moraine cannot be correlated with a younger paleoclimatic event (also see below my comment on lines 439-440, where this correlation was actually done).*

*I suggest that the reversed moraine mean ages should be acknowledged and possible reasons discussed. Are there any field observations that might explain a systematic age underestimation, like enhanced exhumation, erosion, cover on the outer moraines? Exhumation seems unlikely, given the big boulder sizes on the supplement pictures. Could the systematic offset just be a coincidence? It can easily be tested, e.g. by using an MSWD, if the dispersion of all MIF 3 and 4 ages is only due to analytical errors. It could be argued that the moraine formation at the two ice margins occurred within such a short interval that their ages are not distinguishable with the ¹⁰Be dating method. Logically, the average of all ages from the MIF 3 and 4 ice margins would then provide the best estimate of the two glacier stabilizations. Parts of the discussion will need to adjusted to this.*

We thank R1 for the positive evaluation of our manuscript. We agree with R1's main comment, namely that a discussion of the age inversion of moraines JR3/L3 and JR4/L4 – equivalent to Moraine Formation Intervals (MFI) 3 and 4 – is missing from the original version of the manuscript. The revised version includes all of R1's suggestions which address this weakness, specifically:

- adding potential explanations for age inversion of the EH moraines to the discussion *(sections 5.2.1 and 5.2.3)*

- performing a reduced $\chi^2$-test, also known as Mean Squared Weighted Deviation (*MSWD*) statistic *(Supplement, section 4)*

- calculating a mean moraine formation interval based on all boulder ages featured by moraines JR3/L3 and JR4/L4 *(section 5.2.1)*

**2 | MINOR COMMENTS**

**Lines 20-21:** *According to my above comments, better change by saying that two Early Holocene moraine formation intervals occurred around ~11 ka and that they were close both in space and time.*

We agree with R1 and modified our interpretation of the two Early Holocene (EH) moraine formation intervals according to his/her/their suggestion. We combined moraine ages from both valleys to one EH moraine formation interval, yielding an age of 11.0 ±0.7 ka.

*Lines 20–21*

**Line 44:** *revise grammar.*

Revised.

*Lines 44–47*

**Lines 79-80:** *make two sentences*

Sentence has been split.

*Lines 80–81*

**Lines 84:** *Yes, that's indeed important, also for the Late Glacial in the rest of the Alps (just a comment).*

Thank you for supporting our research objective.

**Legend of Fig. 1:** *For the light blue shading that's the ice extent rather than the ice margin. For the national border it might be interesting to indicate to which countries it belongs.*

Legend item "2018 ice margin" was modified to "2018 ice extent"; country labels "AUT" and "CH" were added in Figures 1c and 1d.

*Legend of Fig. 1*
*Lines 105 and 111*

**Line 130:** *something is wrong at the end of this line: "they form" instead of "or"*

We intended to express that moraines form along ice margins that are stable for at least a few years – or longer. We hope that the brackets have improved the semantics of the sentence.

*Lines 133–134*

**Lines 176-179:** *I recommend to at least indicate the potential impact of a possible snow pack scenario, as snow has a higher impact than erosion. The effects of erosion are tested (supplements), but one could argue here as well that it might lead to a higher age dispersion given the various geometries of the boulders. So, testing one but not the other doesn't seem straightforward*

We have included an estimate for the potential impact of a hypothetical snow cover in the revised manuscript version. If applied, boulder ages would become c. 5–6 % older (Gosse and Phillips, 2001).

*Lines 188–190*

**Lines 192-193:** *J1 is dated to the 18th, that's not the end of the LIA. Shouldn't it therefore say that J0's age falls between the LIA maximum and the turn of the 20th century? Or if you just cite Fischer et al. (2019), then phrase the sentence differently ("…is thought to fall…" or similar). The same comment applies to the first line of Fig. 2's caption.*

In our view, the deposition of J1 took not necessarily only place during the 18th century, but probably in the course of multiple glacier advances during the LIA (and potentially before). Our interpretation of the $^{10}$Be exposure age cluster during the 18th century is that during this period,

Jamtalferner certainly reached its LIA maximum. However, there is strong evidence (for instance an historical map, which we refer to in the revised manuscript version) that Jamtalferner was at its maximum again around 1850 CE, which is true for many glaciers in the European Alps. Therefore, we place the age of J0 – located just inside the LIA moraine – into the period between the end of the LIA and the turn of the 20th century.

*Lines 203–205*

**Fig. 3:** *Make sure the same language is used in the text and figures (hut/Hütte)*

Homogenized throughout the manuscript.

**Table 1 and 2:** *The third to last column has a wrong header in each table. The blanks could be added to the tables, notably for easier comparison of the numbers of atoms $^{10}$Be.*

- Column header was corrected.
- We agree with R1 that comparison of $^{10}$Be atoms in samples with $^{10}$Be atoms in blanks would be easier if blanks were added to Tables 1 and 2. However, in the manuscript we present our data valley-wise and based on the landforms corresponding samples were taken from. Process batches in turn were not organized by landforms, but by other criteria (e.g., availability of purified quartz). If we would add blanks to Tables 1 and 2, we would have to list each blank multiple times, which might cause confusion. A batch-wise presentation of all samples is given in the **supplement, Table S2**.

*Tables 1 and 2*

**Lines 308 and 345:** *Given that there's just one sample, I would phrase this more cautiously: "might capture" or similar*

Corrected according to R1's suggestion.

*Lines 347, 387–389*

**Lines 313-315:** *Grouping the L3L moraine segments with the other MIF 3 moraine segments seems arbitrary, because the ages of MIF 3 and 4 are statistically the same. Unless you have more convincing arguments, I would attribute these Left-lateral ridges to both MIF 3 and 4.*

We agree with R1 and changed the name of the ridge "L3L" to "L3-4L" in all figures and the text. In our revision, we used the ages featured by L3-4L (samples LAR-19-22 and LAR-19-24) not only to calculate the landform age L3, but also included them in the calculation of L4 (sh. **Results, section 4.2.2**).

*Figure 4*
*Lines 348–359*

**Fig. 7:** *According to my main comment above, it would make more sense to compare the kernel plots for MIF 3 and 4 in each valley to show that they are indistinguishable from each other, and show their mean ages and insist that they are the same in both valleys.*

Agreed and corrected. Please see also our responses regarding R1's comment to Lines 20-21 and regarding R1's main comment.

*Figure 7e*

**Line 364:** *glacier advance*

Corrected.

*Line 408*

**Fig. 8:** *It would be convenient to have one of the scales in years before present.*

Upper x-axis was changed to "yrs BP".

*Figure 8*

**Lines 345 + 362 + 395-396 etc:** *Are the earlier-LIA and pre-LIA boulders in a stratigraphically different (outer) position compared to the ~18th century boulder? And do you have field evidence that the "LIA" moraines are composite moraines? If yes, it would be good to mention this in the Results – Geomorphology section. Just to interpret this from a few isolated boulder ages is very speculative and should be handle more cautiously.*

The comment that our interpretation is based on a small number of samples, is justified. Therefore, we have formulated the sections addressed by R1 more cautiously. Also, we included the possibility of pre-exposure of the pre-LIA samples.

To the position of early- and pre-LIA boulders (please see also photo documentation in the **supplement, section 6**):

- LAR-19-23 is positioned on the left-lateral LIA moraine (L1) in the Laraintal.
- The left-lateral J1 segment in the Jamtal, where the pre-LIA (neoglacial) boulder **JAM-18-07** was deposited, is not very pronounced, probably because much of the material was eroded due to the steepness of the terrain. The boulder itself is located on the small crest and can clearly be attributed to the J1 moraine.
- **JAM-18-16** (pre-LIA) is deposited on the right-lateral section of J1, in its upper, broader part (Figure 2 in the manuscript), shortly before a large channel. The crest is not clearly defined and hindsight, we believe that this boulder might have toppled.

*Lines 387–389, 407, 439–443*

**Line 396:** *"in the following sections…"*

Corrected.

*Line 440*

**Lines 396-398:** *This sentence probably needs to be revised, it's unclear. I guess you mean that J1 and L1 mark the MAXIMUM glacier advances and temperature minima SINCE the YD-EH transition?*

Proposed changes have been implemented.

*Lines 440–442*

*Revise the title of section 5.2, as it's the same as for section 5.1*

Fixed.

*Lines 443*

**Line 407:** *Nothing is mentioned about the fact that the nominal ages are systematically reversed, see my main comment above.*

Age inversion of moraines L3/JR3 and L4/JR4, albeit statistically non-existent, is discussed in the revised version of the manuscript. We added the following potential explanations:

- Pre-exposure of boulders featured by the inner and older moraine.
- Post-depositional displacement of boulders sampled from the outer and younger moraine.
- Surface erosion of L4/JR4 boulders due to katabatic winds along the L3/JR3 ice margin.

We note that none of the explanations is completely convincing (with more details given in the manuscript). However, the age difference is statistically not significant as highlighted by R1 in his/her/their main comments. Therefore, the age difference does not impede a plausible age interpretation.

*Lines 448–462*

**Line 410:** *Concerning the statement about the climate variability in the studied valleys, is this based on specific observations and does it refer to a specific time period? Can you provide a source?*

This part of the discussion has been moved to the end of **section 5.2.1** and has been formulated more concisely. Also, the statement is now supported with data on summer temperatures from different meteorological stations in the region.

*Lines 531–536*
*Appendix B1*

**Line 411:** *"Also, catchments are comparable…"?*

Corrected.

*Line 531*

**Line 413:** *It's not clear to which "variations in the timing of moraine formation" is referred here. The same applies to "age variability among moraines dated in the region". These variations have not been discussed and cited so far. I guess the statements refer to the Verwall and Ochsental chronologies, but this needs to be clarified.*

This section has been rephrased and clarified.

*Lines 531–536*

**Line 422:** *FIG. 9c-f*

Fixed.

*Line 480*

**Line 432 and 435:** *Were the ELAs in both valleys determined with the same methods? It would be good to give a few more explicit arguments that support the concept that the dated Kartell moraine is not the equivalent of the MIF 3 and 4 moraines and that the age difference is not due to dating uncertainties.*

All reported snowline depression were derived through the Accumulation Area Ratio (AAR) method (Gross et al., 1978), which is now specified in the manuscript.

The regional glacier history concept is mainly supported by numerical age data, and to a lesser extent by ELA depressions reported in previous studies. Uncertainties tied to the dating method complicate the deciphering of distinct glacier advances/stabilizations, specifically the distinction between the Kartell moraines and the MFI 3-4 moraines, and the MFI 3-4 and the Kromer moraines. This problem was briefly addressed in the first version of our manuscript (lines 414–416). In the revised version, we discuss this problem in more detail and point out that moraine ages overlap within uncertainties.

*Lines 527–537*

**Lines 439-440:** *How can MIF 4, being stratigraphically older than MIF 3, be related to a younger paleoclimatic event??? This needs to be corrected. See my main comment.*

Corrected. Please see our response to R1's main comment, and to lines 20-21.

*Lines 493–502*

**Lines 446-337:** *remove "on the one hand" and "on the other hand"*

Removed.

*Lines 507–508*

**Lines 523 and 525:** *Fig. 9 not 8*

Corrected.

*Line 600–601*

**Line 531:** *Fig. 9g-h*

Figure number added.

*Line 613*

**Line 535:** *"which in turn led…"*

Corrected.

*Line 617*

**Line 595:** *Again, a glacier stabilization that is recorded in the most external position cannot have occurred a few centuries later. Please correct this.*

We assume that R1 refers to line 559 given that line 595 does not refer to MFI 3 and/or MFI 4 in the first version of the manuscript. We removed our suggestion that MFI 4 may be linked to cooling between 10.7-10.5 ka as this interpretation is implausible.

*Lines 638–639*

**Line 573:** *this needs to be phrased more cautiously.*

In our revision, we highlight that the data, which supports a glacier advance around 500 CE, is limited in the region.

*Lines 650–653*

**REFERENCES**

Gosse, J.C., Phillips, F.M., 2001. Terrestrial in situ cosmogenic nuclides: theory and application. Quaternary Sci Rev 20, 1475-1560.

Gross, G., Kerschner, H., Patzelt, G., 1978. Methodische Untersuchungen über die Schneegrenze in den alpinen Gletschergebieten. Zeitschrift für Gletscherkunde und Glaziogeologie 12, 223-251.

---

## Author Comment (AC2)

**Title:**  **Early Holocene cold snaps and their expression in the moraine record of the Eastern European Alps**

**Authors:**  **Sandra M. Braumann, Joerg M. Schaefer, Stephanie M. Neuhuber, Christopher Lüthgens, Alan J. Hidy, Markus Fiebig**

**Article ID:**  **cp-2021-54**

The thank Reviewer #2 (R2) for reviewing the manuscript, and for his/her/their positive feedback. All comments and suggestions of R2, which helped to improve the manuscript, are included in the revised version of the manuscript and are point by point commented in the following document. R2's comments are listed in the gray boxes followed by our responses in blue.

**1 | M A I N   C O M M E N T S**

**Summary of the manuscript**
*This manuscript presents 27 new Be-10 dates on the timing of Holocene glacial advances in the Eastern Alps of Europe and correlates them with inferred meltwater events within the North Atlantic. Their findings corroborate previous studies from the Western Alps and peripheral regions of the North Atlantic, which is glacial advances in those regions were the result of cold atmospheric temperatures being advected eastward from the Atlantic, and that the Atlantic was cold as a result of a slowdown in AMOC, which was a result of enhanced freshwater input from the Laurentide Ice Sheet.*

**Summary of my assessment**
*This manuscript was an absolute delight to read! Scientifically, the results support the conclusions (with a few very minor exceptions, discussed below) and the discussion was beyond interesting, it was enlightening. Aesthetically, the writing was silken, the figures were beautiful, and the logical organization of the manuscript was thoroughly evident. In my opinion, this paper should be published with a few moderate and several minor revisions.*

**Major Points**
*As I said, I have no "major" issues with this paper. I do, however, have three "moderate" points for the authors to consider:*

***First***, *there is a slight age inversion between MFI 4 (dated to 10.8 ± 0.7 ka) and MFI 3 (dated to 11.2 ± 0.8 ka). While these two events are indistinguishable in timing from each other based on the Be-10 dates, we know from the geomorphology that MFI 4 must be older than MFI 3. While this apparent age inversion is noted on line 407, it isn't (as far as I noticed) mentioned again. Thus, the discussion on lines 437-441 seems strange. The authors say MFI 3 falls within the PBO (defined as 11.30-11.15 ka in Europe), but MFI 4 does not. In reality, MFI 4 must have preceded MFI 3, so if only one of them is associated with the PBO, it's more likely MFI 4. In short, I don't see how both of the following could be true: (1) MFI 3 correlates with the PBO and (2) MFI 4 postdates the PBO and correlates with a summer cooling detected in Swiss and Austrian lake sediments.*

***Second***, *I was surprised that the authors did not acknowledge the possibility that the boulders dating to c. 700-1500 years ago (JAM-18-07, JAM-18-16, and LAR-19-23) were actually deposited more recently (c. the 18th century?) and contain inherited nuclides from prior exposure. The evidence from these two valleys for glacial advances c. 500 CE and c. 1300 CE is tenuous, in my opinion. The data permit glacial advances at those times, but the evidence is not compelling. I was pretty skeptical of glacial advances at those times in these two drainages—until I read about the evidence elsewhere in the Alps for glacial advances at those times (e.g., the boulder from Ochsental dating to 1500 ± 40 years, the sediment and peat profiles from various glacial forefields in the Eastern Alps, and the documented glacial advances in the Western Alps at those times. In light of the evidence from elsewhere in the Alps for glacial advances at c. 500 CE and 1300 CE, I think the interpretation presented by the authors—that these two valleys also hosted ice advances at those times—is reasonable, but the alternative possibility, that these three samples contain inherited nuclides from prior exposure should be discussed in the text.*

***Third*** *(and perhaps less importantly than the two previous points), I find it curious that both valleys have moraine ridges just outside the LIA margins that are ~8-10 m wide, rich in fine-grained sediment, and devoid of boulders. While the authors suggest these ridges (J2 and L2) might be equivalent to the c. 10 ka Grüne Kuppe moraine in the adjacent landscape of Ochsental (lines 452-465), the fine-grained nature of these ridges seems anomalous—and suggests to me that they have a different origin from the boulder-rich ridges (moraines) present in these alpine valleys. In particular, I wonder if J2 and L2 might be some sort of push-ridge associated with the LIA advance? That's speculative, of course, but no more speculative (in my opinion) than their potential association with the Grüne Kuppe moraine. In any event, the true age of the undated J2 and L2 ridges is not an essential point to this manuscript—but their fine-grained nature does seem anomalous and cry out for an explanation.*

We thank R2 for the appreciation of our manuscript. We acknowledge the three main points raised by R2 and improved corresponding sections in the revised version of the manuscript.

1. The age inversion of moraines JR3/L3 and JR4/L4 – equivalent to Moraine Formation Intervals (MFI) 3 and 4 – is discussed in the updated manuscript version: We agree with both reviewers that the timing of MFI 3 and MFI 4 is statistically indistinguishable. Therefore, we followed R1's suggestion and combined boulder ages featured by corresponding landforms and calculated a combined MFI 3-4 (11.0 ±0.7 ka). This phase of moraine stabilization is then correlated with climatic events during the Early Holocene (EH) *(sections 5.2.1 and 5.2.3)*. We believe that this strategy has improved the plausibility of corresponding sections in the discussion.

2. All text passages in which we propose a potential early Little Ice Age (LIA) advance around 1300 CE are formulated more cautiously in the revised version, since evidence is limited to one single boulder age (LAR-19-23). Furthermore, we revised section 5.1 and now discuss the possibility of pre-exposure in the context of Neoglacial boulder ages **(section 5.1)**.

3. We highlighted the fine-grained texture of J2 and L2 compared to the blocky EH moraines *(section 4.2.2)* and suggested different ice dynamics (advance vs. equilibrium) being responsible for this difference. In the Discussion (**section 5.2.1**), we corrected our age estimate by including the LIA period.

**2 | MINOR COMMENTS**

**Line 16:** *I think there should be a hyphen between ice and margin.*

Hyphen added.

*Line 16*

**Lines 20-21:** *I think the MFI should be listed in chronological order, so oldest first.*

The strategy to constrain Early Holocene (EH) moraine formation in the study area has been improved in the updated version. As boulder ages that are featured by EH moraines in both valleys are statistically indistinguishable, they have been merged and used to calculate a combined MFI, yielding an age of 11.0 ±0.7 ka (see also response to ***Main comments – 1.***)

*Lines 20–21*

**Line 24:** *I think "millennial scale" should be hyphenated.*

Corrected.

*Lines 24–25*

**Line 25:** *I think "contemporaneous" is possibly too strong; I'd suggest "indistinguishable in timing" instead.*

Replaced with "coincide".

*Line 25*

**Line 32:** *I think a reference or two should be provided for the duration of the YD and (less importantly) for the beginning of the Holocene. Yes, these dates are common knowledge—but someone(s) did some groundbreaking research at some point to determine those dates and their efforts should be acknowledged, in my opinion.*

References added.

*Lines 32–33*

**Line 35:** *"Centennial scale" should always be linked by a hyphen, I believe, so: "centennial-scale."*

Fixed throughout the manuscript.

**Line 35:** *I remember Bob Anderson (CU-Boulder) commenting on one of my manuscripts and saying that "to be" was the weakest of all verbs, so to avoid whenever possible. In this case, "to be" can be deleted and the sentence remains grammatically correct (and becomes more concise).*

Corrected.

*Line 36*

**Line 36:** *Stylistically, I think references should also be cited in chronological order, because that helps me learn the history of the field, but I recognize that some journals prefer/require citations to be in alphabetical order. In this case, it's neither. Also, the "e.g." should be followed by a comma (so, "e.g..,") and there should be a space after the semicolons and before the next name.*

The citation style template of "Climate of the Past", which to our knowledge arranges authors alphabetically, is used in the revised manuscript version. "e.g." has been changed to "e.g.," throughout the text.

**Line 77:** *"State of the art" should be hyphenated, I believe. So, "state-of-the-art."*

Corrected.

*Line 78*

**Figure 1:** *Is an absolutely stunning figure, it contains so much information, and is an absolute delight to look at. I hope the person (or people) who made it are proud of their efforts, it is a real accomplishment.*

We are grateful for positive feedback!

**Line 101:** *"Geographic" can be deleted. (Are their locations that are not geographic? I can't think of any… so the word can be deleted for greater conciseness without loss of meaning.*

Removed.

*Line 103*

**Line 112:** *Delete the "a" before "snow cover."*

Deleted.

*Line 115*

**Line 126:** *This might be the previous semester speaking—in which I taught mineralogy/petrology despite being a geomorphologist—but I think "quartz yield" would be a better phrase here than "quartz content." The lithologies present in these valleys must truly have greater quartz "contents" than these, as some of the quartz was etched away in the cleaning process, was it not? "Quartz content," to me, suggests some sort of modal norm, potentially applicable to some sort of classification scheme—while "quartz yield" suggests this is how much quartz we got out of the rock.*

Corrected.

*Line 129–130*

**Line 134:** *How do you know the boulders were carved out of the bedrock by glacial flow? Why couldn't they have tumbled onto the glacier's surface from the adjoining cliffs and been carried to the ice margin. In theory, the boulders were plucked out of the bedrock and never saw "the light of day" until they were deposited on the moraines—but was this really what happened? How do we know?*

When possible, we aimed for (sub-)rounded boulders to maximize the probability of sub- and englacial transport. A more detailed description has been added to the paragraph.

*Lines 137–140*

**Line 142:** *Is there a difference between "complemented" and "updated?" If not, I'd delete complemented and just use updated. I think "updated" is the more commonly used verb.*

Deleted.

*Lines 147*

**Line 144:** *I think the first "historical" should be deleted. First, I'm not sure those moraines are really historical, in the meaning of the word, and second, the sentence is somewhat circular sounding as written.*

Sentence modified according to R2's suggestions.

*Lines 149-150*

**Lines 150-151:** *Is the datum for this DEM actually sea level, and not some ellipsoid height? (I know this is nitty-gritty technical, but would you please confirm which the DEM is referencing with regards to its elevation?) See the last two paragraphs of Greg Balco's blog post here (https://cosmognosis.wordpress.com/2017/03/28/is-a-cheap-gps-ok-for-elevation-measurements-or-do-you-need-a-fancy-one/) for more details, but the gist is that this is a frequent and impactful error on cosmo ages.*

Altitudes derived from the Digital Elevation Model (DEM) refer to sea level. More details can be found at the following link, which unfortunately is not translated into English:

https://www.bev.gv.at/portal/page?_pageid=713,1572984&_dad=portal&_schema=PORTAL

"Höhenbezug" specifies the elevation reference system, which in the case of the used DEM is "Adria Triest".

**Line 157:** *As above, I think "quartz content" should be "quartz yield."*

Corrected.

*Line 163*

**Line 171:** *How can an individual boulder have an arithmetic mean age? (Other than by the counting statistics at the accelerator, which I'm assuming is not being referenced.) If the authors are referring to the counting statistics from the accelerator's beam line (or whatever it's called), that should be specified for clarity. My hunch, though, is that this statement (of the boulder ages being arithmetic mean ages) is a typo—and that only the landform ages are arithmetic means. But please, if I'm off the mark with my hunch, just clarify what's being discussed/reported here.*

The expression is indeed confusing. In fact, we intended to refer to the mean of replicate AMS measurements and have revised the section accordingly.

*Lines 177–178*

**Line 186:** *Reviewers don't often say this (at least I don't think they do…), but I just want to praise how logical and organized the Results and Discussion are, the thought that the authors have put into this manuscript is clearly evident!*

Thank you for acknowledging our efforts to present our results in a comprehensible and easy-to-read manner.

**Line 198:** *As above, I think the fine-grained nature of ridges J2 and L2 is an important observation with regards to their origin(s).*

See our response in the **Main comments, point 3**.

**Figure 2:** *Is absolutely stunning… I really like how the authors have shaded the moraines by color and used the right-angle lines to connect the sample dots with the boxes with their ages and names. The 1-m hillshade background doesn't hurt either!*
**Figure 3:** *Also very nice!*

Thank you for appreciating the design of our figures!

**Line 244:** *The abbreviation "c" shouldn't be used to start a sentence, even in capitalized form. Instead, I'd recommend using "approximately" instead. Also, in my experience "c." or "ca." for "circa" are only used with regards to approximate dates, not approximate distances, or other quantities.*

Corrected throughout the manuscript.

**Figure 4:** *Awesome, just like Figure 2…*
**Figure 5:** *Beautiful, just like Figure 3.*

Thank you!

**Line 276:** *I think the word "risky" is misleading. At first I thought you were saying that it was too dangerous to sample there, as rocks might fall down as squish you while you're trying to hammer out a sample, but then the phrase "to tackle with SED of boulders" made me realize that you're talking about scientific risk, not personal risk. I'd encourage you to preface risky with scientific, so "scientifically risky," unless you are concerned about being squished, in which case I'd recommend you drop the "to tackle with SED of boulders" part of the sentence.*

Both, actually, but we are more concerned about producing erroneous data. Corrected accordingly.

*Line 311*
* * *
**Line 293:** *I'm not seeing the gray-shaded bars… do you mean the thin black lines instead, for each individual sample? Perhaps "bars" should be singular, "bar," instead? Also, the vertical uncertainty bar in the background—which I'm guessing is what you're talking about—seems kind of greenish, but perhaps that's me.*

The word "bars" has been changed to the singular form ("bar"). The gray shading indicating the 1σ analytical uncertainty has been slightly darkened. Further explanations of the color codes in the figure and their meaning have been added to the captions.

*Line 328*
* * *
**Figure 6:** *Also, while we're talking about Fig. 6, I don't really understand the differences between the three types of uncertainty you report on the mean age, and I think other readers might also struggle. I understand total uncertainty (reported in red) is the production rate and spike uncertainty added in quadrature, but I don't really understand the importance of the difference between the 1-sigma uncertainty (±16 years) and the standard error of the mean (±9). I guess I would have thought they were the same thing, with the "standard error of the mean" being the correct term and the "1-sigma uncertainty" being the not-really-100%-approved-by-statisticians version of the "standard error of the mean." Your paper isn't the place to teach readers about statistics—but a few sentences somewhere in the text about the differences between these things would help me (and I think others) out. (Thanks for considering this request!)*

Thank you for pointing out the lack of clarity in the captions. We have added some information on the Standard Error to the **captions of Figure 6**. Standard Deviation (SD) and Standard Error (SE) are related as the SE is calculated based on the SD, but are different statistical measures.

The **Standard Deviation (SD)** describes the deviation of the data relative to their mean and is calculated using the following formula:

$$SD = \sqrt{\frac{\sum(x - \bar{x})^2}{n - 1}}$$

*x … boulder ages*
*$\bar{x}$ … mean of boulder ages*
*n … number of boulders*

The **Standard Error (SE)** quantifies the dispersion of sample means around the "true" population mean. It describes potential sampling fluctuation of a distribution and was calculated using the following formula:

$$SE = \frac{SD}{\sqrt{n}}$$

For clarification:

- Individual **boulder ages** are reported with 1σ SD of AMS measurements.
- Uncertainties reported with **moraine ages** are: 1σ SD of AMS measurements, a 1 % uncertainty on the carrier concentration, and the uncertainty of the production rate propagated in quadrature and are very conservative (given a 6.3 % uncertainty on the Swiss production rate).
- In the Kernel plots, we report the Standard Error of moraine ages to give the reader a sense for dispersion of sample means around the population's mean.

*Lines 329–333 and 380–383*
* * *
**Line 300:** *I think "supplements" should be capitalized, and the final "s" in it dropped. So, "Supplement."*

Corrected throughout the manuscript.

**Table 1:** *Are the elevations really meters above sea level, or meters about some ellipsoid? (As above, what is really 0 m elevation in the DEM?) Also, "quartz weight" would be better reported as "quartz mass." Also, I think the 10th column from the left (3rd from the right) is mislabeled. It says it's a ratio of Be-10 to Be-9, but that's what the column to the left reports and the units in this column are atoms, and a ratio shouldn't have units. Finally, how can we possibly know the average thickness of the samples to a 10th of a millimeter? While you could just reduce the reported precision (as I suspect the true precision of these measurements is being overstated)—I'm also curious as to what method you used to calculate these numbers, as it's not stated in the text (that I saw) and I'd potentially like to use the same method in my own research.*

- Elevation: altitudes refer to sea level (please see our response to lines 150-151).
- Column header "Quartz weight" has been changed to "Quartz mass".
- Column header "$^{10}Be/^{9}Be$ ratio" has been changed to "10Be atoms".
- Average sample thickness: It is correct that the specification of the second decimal place pretends "false" precision. Therefore, one decimal place was removed. The average thickness was measured with a caliper. For each sample, multiple thickness measurements were made at various points (roughly based on a 2x2 cm grid). Then, an average value was calculated.

*Table 1*

**Line 313:** *"11 230" should be reported as "11,230" or, alternatively, all the other 5-digit ages should be reported in the same fashion, with a space between the 3rd and 4th digits.*

Fixed.

*Line 353*

**Lines 323-327:** *This advancement—of adding a late spike of Fe to help process small-volume samples—seems worth of a sentence in the abstract, in my opinion. It's a really methodological advancement for cosmo, I think, and well worth highlighting, in my assessment.*

We appreciate this, however, our manuscript is an example for the application of the Fe addition, which allows for reducing the amount of Be carrier added to the sample. Here, we focus more on the geochronological and climatic interpretation of the data, and less on methodological aspects. We feel that mentioning in the abstract as a highlight would mandate further explanation beyond the scope of this study to include the series of experiments and datasets that were the basis for adopting this technique. A methodology manuscript, in which the development and application of this advancement is described in detail is currently in preparation.

**Table 2:** *Same 4 points as for Table 1 above.*

Fixed.

*Table 2*

**Figure 7:** *Same comment about the meaning of the reported statistics (i.e., what's the difference between the 1 sigma uncertainty and the standard error of the mean) as for Figure 6. In other words (in case my point/uncertainty has been unclear), what are these two statistical calculations telling us? When would/should we refer to one and when should we refer to the other?*

Please see our response to Figure 6.

**Lines 344-345:** *I think 1-3 references would be appropriate here. Who did the work that leads us to expect LIA moraine formation between 1250 and 1850 CE?*

References added.

*Line 389*

**Line 345**: *As discussed above in the "moderate points" section, I think "recorded" is too strong a word for the evidence this manuscript presents, "suggested" might be a good alternative.*

Reviewer suggestion adopted.

*Lines 387–389*

**Line 361**: *While you interpret JAM-18-16's 1070 yr age as a minimum age, because that dates to the beginning of the MWP and the boulder might have toppled. The boulder might, alternatively, host inherited nuclides, in which case the 1070 yr age would be a maximum age. I think this possibility should be acknowledged in the text.*

R2 raises an important point as the boulder's nuclide inventory could indeed contain an inherited component, which would lead to an age overestimation. However, as stated in the manuscript text, we revisited the boulder and argue that due to its bedding and the size of the boulder, it is much more likely that it toppled. Nevertheless, we cannot rule out pre-exposure and have added this explanation to the revised version of our manuscript.

*Line 407*

**Line 377**: *"Centennial scale" should be hyphenated.*

Corrected throughout the manuscript.

**Figure 8**: *There isn't a year 0 in the CE chronologic system, as far as I know. I believe the "CE" dating convention is the same as the "AD" dating convention, with the years matching one-to-one and the meaning ascribed to them (the two-letter abbreviation) being the only difference. Because the Roman's didn't have the concept of zero, the numbering goes directly from 1 BC to 1 AD. Thus, as far as I know, the CE system also goes from -1 to 1 with no zero (but feel free provide a citation correcting me if you think I'm wrong). Also, on a much less nit-picky note, Figure 8 is really well designed! I like your use of color—and how time runs from left to right, that makes it easier to read in my opinion.*

R2 is correct that the CE dating convention does not include the year 0. Therefore, we have modified the x-axis following the example of one of the publications cited in this diagram (Büntgen et al., 2011).

*Figure 8*

**Line 399**: *This is the same section heading as 5.1. I think you might mean "The moraine record of the early Holocene" here.*

Fixed.

*Line 443*

**Line 462**: *I don't think the word "evolving" should be used with regards to species, except in the case of evolution in the Darwinian sense. I think you might be thinking of an upward migration in the treeline, is that right? Also, pinus cembra should be italicized, I think, and shouldn't pinus be capitalized as well?*

Rephrased and corrected.

*Line 523*

**Figure 9**: *I'm not familiar with the word "warves." It's at the top of the figure, sideways, in association with PBO. Do you mean varves instead? And if so, what do varves have to do with the PBO? Also, with regards to (g) and (h), what do the uncertainty bars represent, 1-sigma uncertainties?*

"Warves" is a misspelling; we meant "varves". The snippet is a remnant of a first draft of the figure and has been removed from the revised version.

*Figure 9*

**Line 467**: *I'm not familiar with the word "detrical." Do you mean "detrital" instead?*

Typo. Corrected.

*Line 543*

**Line 491**: *I had a stats prof who said the word "significant" should only be used in scientific writing to mean "statistically significant" (at a specified threshold). That might be a stats professor's view of the world—but I see no reason why "substantial" couldn't be used here instead.*

Replaced.

*Line 567*

**Line 491**: *"large scale" should be hyphenated, I believe.*

Hyphen added.

*Line 567*

**Line 493**: *Was the Icelandic ice mass really an ice sheet (>50,000 km2) vs. an ice cap (<50,000 km2)?*

According to Geirsdottir et al. (2009, Fig. 2c), large parts of Iceland (total area 103,000 km$^2$), were still covered by glacial ice during the Preboreal. Therefore, we think that the term ice sheet is adequate in this context.

**Line 501**: *I believe "hemisphere" should be capitalized.*

**Line 502**: *Essentially the same point, I think "northern hemisphere" should be capitalized.*

Corrected throughout the manuscript.

**Line 508-509**: *I have two minor points about "there is evidence of more subdued glacial discharge during the EH that results in a deceleration of the thermohaline circulation." First, I think 1-3 references should be cited here, in support of that statement. Second, the term "thermohaline circulation" seems to have gone out of fashion, so I'd recommend you use "AMOC," meridional overturning circulation, or something of that nature.*

As suggested by R2, we added three references to support our statement. Regarding the second point, we prefer to keep the term "thermohaline circulation", as we would like to highlight the physical process itself, not so much the specific ocean current.

*Lines 586–587*

**Line 513**: *I think "glaciated" should be deleted, because it is redundant with "glaciers" two words previous.*

Deleted.

*Line 591*

**Line 517**: *"ice bergs" should be one word, "icebergs."*

Corrected.

*Line 595*

**Line 518**: *I don't know if you need to introduce the term/concept "H0" into the paper. Its fine if you want to—but, if I'm not mistaken, the term is only used in this paragraph (lines 515-531) and "H0" does not appear on Fig. 9 (YD is used there instead).*

We prefer to keep the explanation of the term as a service to the reader.

**Line 519:** *I'm not familiar with the word "detrical," do you mean "detrital" instead?*

Typo – corrected.

*Line 597*

**Line 521-522:** *The sentence says Jennings et al. (2015) found 6 DCPs, but Fig. 9 shows 7. Maybe someone discovered a 7th one later (I don't know that literature well enough), but it seems like possibly a typo so I thought I'd mention it.*

Jennings et al. (2015) found 7 major and one minor DCP between 11.5 and 8.15 ka. In Fig. 9, we only highlight the major ones, hence 7. "DCP 1-6" was indeed a typo.

*Line 600*

**Line 523:** *The reference to "Fig. 8a" should be to Fig. 9a.*

Fixed.

*Lines 600–601*

**Line 531:** *A figure number is missing from "Fig. g-h."*

Figure number added.

*Line 610*

**Line 534:** *I think "Resulting" should be preceded by "The," so "The resulting freshwater…"*

Corrected.

*Line 613*

**Line 543:** *"sea ice" should be hyphenated.*

Hyphen added.

*Line 622*

**Line 555:** *I think there's a typo here… do you mean the western margin of the North Atlantic? Or the eastern margin of the LIS? The "western" margin of the LIS would be in the Canadian prairie... at least, that's the thought that comes to my mind as I read the sentence.*

We meant the eastern margin of the LIS and thank R2 for this comment.

*Line 634*

**Line 556:** *I think "perturbated" should be replaced with "reduced," as "perturbed" (note the spelling difference) means changed or altered, while "reduced" specifies the direction of change.*

Replaced.

*Line 636*

**Line 573:** *Delete "have," it's not necessary.*

Sentence rephrased.

*Lines 654–655*

**Appendix A:** *I think it's great that you included so many photos of the sampled landscapes, that really helps the curious reader understand each boulder's context within the landscape.*

Thank you!

**Line 612:** *"unweatherd" should be "unweathered."*

Corrected.

**Line 617:** *I looked for your samples on the ICE-D Alpine database and was unable to find them. What's the timeline for getting the samples posted?*

We will have the data posted on the ICE-D database as soon as the manuscript and thus our data organization and data interpretation are accepted.

**Line 632:** *Is "inatura" supposed to be capitalized? (I don't know… but it seems likely to me.)*

"inatura" is written in lower case according to their website (https://www.inatura.at/en/legal-disclosure).

**The Supplement:** *Once again, it's really nice that you included multiple photos of every boulder, I like your thoroughness!*

*Supplement, 5th line to the caption for Table A2: I think the "therefore" in "and was therefore used to quantify" should be deleted, its unnecessary (the sentence makes sense without it).*

*One final note: thanks so much for submitting this manuscript to Climate of the Past. I know reviewers don't often tack on such comments—but your manuscript was such a delight to read, and the figures were so well made!*

The adverb "therefore" was removed in the **captions of Table S2**.

We thank R2 for his/her/their nice and motivating words! We are pleased that the manuscript has been well received and that we have been able to contribute to the advancement of knowledge on glacier and climate fluctuations during the Late Glacial and during the Holocene.

**REFERENCES**

Büntgen, U., Tegel, W., Nicolussi, K., McCormick, M., Frank, D., Trouet, V., Kaplan, J.O., Herzig, F., Heussner, K.U., Wanner, H., Luterbacher, J., Esper, J., 2011. 2500 Years of European Climate Variability and Human Susceptibility. Science 331, 578-582.

Geirsdottir, A., Miller, G.H., Axford, Y., Olafsdottir, S., 2009. Holocene and latest Pleistocene climate and glacier fluctuations in Iceland. Quaternary Sci Rev 28, 2107-2118.

Jennings, A., Andrews, J., Pearce, C., Wilson, L., Olfasdotttir, S., 2015. Detrital carbonate peaks on the Labrador shelf, a 13-7 ka template for freshwater forcing from the Hudson Strait outlet of the Laurentide Ice Sheet into the subpolar gyre. Quaternary Sci Rev 107, 62-80.